# Lifespan-extending interventions induce consistent patterns of fatty acid oxidation in mouse livers

Kengo Watanabe [1], Tomasz Wilmanski[1], Priyanka Baloni[2], Max Robinson[1], Gonzalo G. Garcia[3], Michael R. Hoopmann[1], Mukul K. Midha [1], David H. Baxter[1], Michal Maes[1], Seamus R. Morrone[1], Kelly M. Crebs[1], Charu Kapil [1], Ulrike Kusebauch[1], Jack Wiedrick[4], Jodi Lapidus[4], Lance Pflieger[1,5], Christopher Lausted [1], Jared C. Roach[1], Gwênlyn Glusman [1], Steven R. Cummings [6,7], Nicholas J. Schork[8,9], Nathan D. Price[1,10,11,12], Leroy Hood[1,5,11,12,13✉], Richard A. Miller [3,14], Robert L. Moritz [1] & Noa Rappaport [1✉]

Aging manifests as progressive deteriorations in homeostasis, requiring systems-level perspectives to investigate the gradual molecular dysregulation of underlying biological processes. Here, we report systemic changes in the molecular regulation of biological processes under multiple lifespan-extending interventions. Differential Rank Conservation (DIRAC) analyses of mouse liver proteomics and transcriptomics data show that mechanistically distinct lifespan-extending interventions (acarbose, 17α-estradiol, rapamycin, and calorie restriction) generally tighten the regulation of biological modules. These tightening patterns are similar across the interventions, particularly in processes such as fatty acid oxidation, immune response, and stress response. Differences in DIRAC patterns between proteins and transcripts highlight specific modules which may be tightened via augmented cap-independent translation. Moreover, the systemic shifts in fatty acid metabolism are supported through integrated analysis of liver transcriptomics data with a mouse genome-scale metabolic model. Our findings highlight the power of systems-level approaches for identifying and characterizing the biological processes involved in aging and longevity.

[1] Institute for Systems Biology, Seattle, WA, USA. [2] School of Health Sciences, Purdue University, West Lafayette, IN, USA. [3] Department of Pathology, University of Michigan School of Medicine, Ann Arbor, MI, USA. [4] Oregon Health and Science University, Portland, OR, USA. [5] Phenome Health, Seattle, WA, USA. [6] San Francisco Coordinating Center, California Pacific Medical Center Research Institute, San Francisco, CA, USA. [7] Department of Epidemiology and Biostatistics, University of California, San Francisco, CA, USA. [8] Department of Quantitative Medicine, The Translational Genomics Research Institute (TGen), Phoenix, AZ, USA. [9] Department of Population Sciences and Molecular and Cell Biology, The City of Hope National Medical Center, Duarte, CA, USA. [10] Thorne HealthTech, New York, NY, USA. [11] Department of Bioengineering, University of Washington, Seattle, WA, USA. [12] Paul G. Allen School of Computer Science & Engineering, University of Washington, Seattle, WA, USA. [13] Department of Immunology, University of Washington, Seattle, WA, USA. [14] University of Michigan Geriatrics Center, Ann Arbor, MI, USA. ✉email: lee.hood@isbscience.org; noa.rappaport@isbscience.org

Aging manifests as progressive impairments in maintenance of both cellular and systemic steady states. In humans, it is accompanied by increased risks for chronic conditions such as diabetes, heart disease, neurodegeneration, and cancer[1,2]. Interventions targeting aging mechanisms could delay or moderate chronic diseases and improve health and lifespan[3]. However, aging involves diverse and interconnected molecular and physiological components, posing a challenge for comprehensive understanding[4,5]. For instance, many studies have demonstrated key roles for nutrient-sensing pathways in aging and longevity across species, including growth hormone (GH) and insulin/insulin growth factor 1 (IGF-1), AMP-activated protein kinase (AMPK), sirtuins, and mammalian (or mechanistic) target of rapamycin (mTOR) signaling pathways[6–10], but these nutrient-sensing pathways are intricately interconnected with each other. In addition, the roles of these pathways are not necessarily consistent among tissues, as shown by tissue-specific knockout mice[11,12]. Given the complex and multifaceted nature of aging, systems-level approaches may provide complementary perspectives to single molecule-level approaches and deepen our understanding of the aging processes.

Some nutritional and pharmacological interventions consistently extend lifespan and healthspan (i.e., the period free from age-associated diseases and disabilities)[13] in mouse and other animal models[3,14–16]. Nutritional interventions include calorie restriction (CR)[17], methionine restriction[18], and ketogenic diet[19,20]. While the number of possible geroprotectors (i.e., drugs aiming to prevent, slow, or reverse aging process)[21] has been growing, pharmacological interventions whose effects on lifespan extension have been documented by the National Institute on Aging (NIA) Interventions Testing Program (ITP)[22] include acarbose (ACA)[23–25], canagliflozin[26], 17α-estradiol (17aE2)[23,24,27], glycine[28], nordihydroguaiaretic acid[23,24,29], Protandim® (a Nrf2 inducer)[24], and rapamycin (Rapa)[30–32]. Rapa is a macrolide compound found to prolong lifespan in every organism studied, including yeast, worms, flies, and mammals[33,34]. Rapa modulates nutrient-sensing pathways by inhibiting the activity of mTOR through complex formation with FK506-binding protein 12, which globally attenuates protein translation via mTOR complex 1 (mTORC1) and ultimately reduces inflammation, increases autophagy, and improves stem cell maintenance[35,36]. ACA could share some aspects of CR[23]; it is an oral antidiabetic drug which competitively inhibits the activity of α-glucosidase enzymes to digest polysaccharides, resulting in the delay of sugar uptake in the gastrointestinal tract[37]. ACA treatment has been shown to extend lifespan in male mice more than in female mice[23–25], possibly related to sex-dependent differences observed in heart, liver, and gut metabolite profiles[38,39]. 17aE2 is a stereoisomer of the dominant female sex hormone 17β-estradiol, having much weaker binding affinity to the classical estrogen receptors, stronger affinity to the brain estrogen receptor, and neuroprotective properties[40,41]. 17aE2 treatment extends lifespan in male but not in female mice[23,24,27], potentially related to male-specific reduction of age-associated neuroinflammation[42] and sex-specific metabolomic responses observed in liver and plasma metabolite profiles[43]. Because these lifespan-extending drugs were tested with standardized protocols in the NIA ITP and because they have differences in primary mode of action, comparisons of their effects on molecular regulation are valuable for our understanding of common, fundamental, or core aging and longevity mechanisms.

A module of a biological system can be represented as a molecular network where nodes and edges correspond to biomolecules (e.g., gene transcripts, proteins, and metabolites) and relationships (e.g., physical interactions, chemical reactions), respectively. For each sample, ranks of biomolecules can be obtained from experimental data by ordering the values of interest (e.g., abundances, levels of specific post-translational modification) between the biomolecules within a module (Fig. 1a). When these ranks are highly conserved among the samples within a population of a specific phenotype, the module is considered tightly regulated in the population, because biological regulatory mechanisms or pressures must act consistently across the samples to produce this high conservation pattern. In contrast, low rank conservation among the samples within a phenotype indicates loose module regulation in the population. The Differential Rank Conservation (DIRAC) method[44] can quantify this population-level variability for a given biological module (i.e., a biomolecule set, typically defined with an a priori gene network or pathway) by utilizing differential ranks of the pairwise rank comparisons, instead of absolute ranks, within the module (Fig. 1b). A previous study applied DIRAC analysis to gene expression data of multiple cancer types, and revealed general loosening of BioCarta-defined modules in more malignant phenotypes and later stages of disease progression[44], indicating that a loss of tight regulation characterizes the dysregulation of biological processes in cancer. Hence, the DIRAC method can be used for identifying biological modules whose regulatory patterns are changed by lifespan-extending interventions.

In this study, we report systemic changes in the molecular regulation of biological processes under multiple lifespan-extending interventions, by jointly leveraging systems-level analyses on two mouse liver proteomic datasets, which were generated in the NIA Longevity Consortium, and a previously published mouse liver transcriptomic dataset[45]. We apply DIRAC analysis to the protein abundance profile, first with a set of predefined modules derived from Gene Ontology Biological Process (GOBP) annotations and then with a set of unbiased modules derived from Weighted Gene Co-expression Network Analysis (WGCNA)[46,47], and demonstrate that three lifespan-extending drugs (ACA, 17aE2, and Rapa) promoted tighter regulation of biological modules, such as fatty acid oxidation, immune response, and stress response processes. Moreover, DIRAC pattern comparisons between proteomics and transcriptomics data suggest that the lifespan-extending interventions tightly regulated biological modules at different levels, including post-transcriptional alterations through cap-independent translation (CIT) of specific mRNAs[48]. As a complementary approach, mouse genome-scale metabolic model (GEM)[49,50] reconstruction with the transcript abundance profile supports that multiple lifespan-extending interventions shifted fatty acid metabolism.

## Results

**Lifespan-extending interventions increased proteomic profile conservation in a priori modules.** To compare the systems-level changes induced by different lifespan-extending interventions, we first applied DIRAC analysis to a liver proteomic dataset which was generated through a mouse lifespan-extending intervention experiment in the NIA Longevity Consortium (denoted LC-M001 proteomics). In this LC-M001 experiment, 48 mice were either untreated (Control) or subjected to one of three lifespan-extending drug treatments (ACA, 17aE2, or Rapa), and euthanized at 12 months ($n = 12$ (6 female and 6 male) mice per group). The design of evaluating drug effects on healthy young adult mice was motivated by the desire to reduce confounding effects of aging and of late-life diseases. Acknowledging the limited sample size for statistical power and robustness of downstream analyses, we regressed out the potential effects of sex in advance (see Methods) and pooled female and male samples per intervention throughout the current study (see Discussion). As biological modules, we defined 164 a priori modules by mapping the measured proteins onto GOBP annotations (see Methods; Supplementary Data 1).

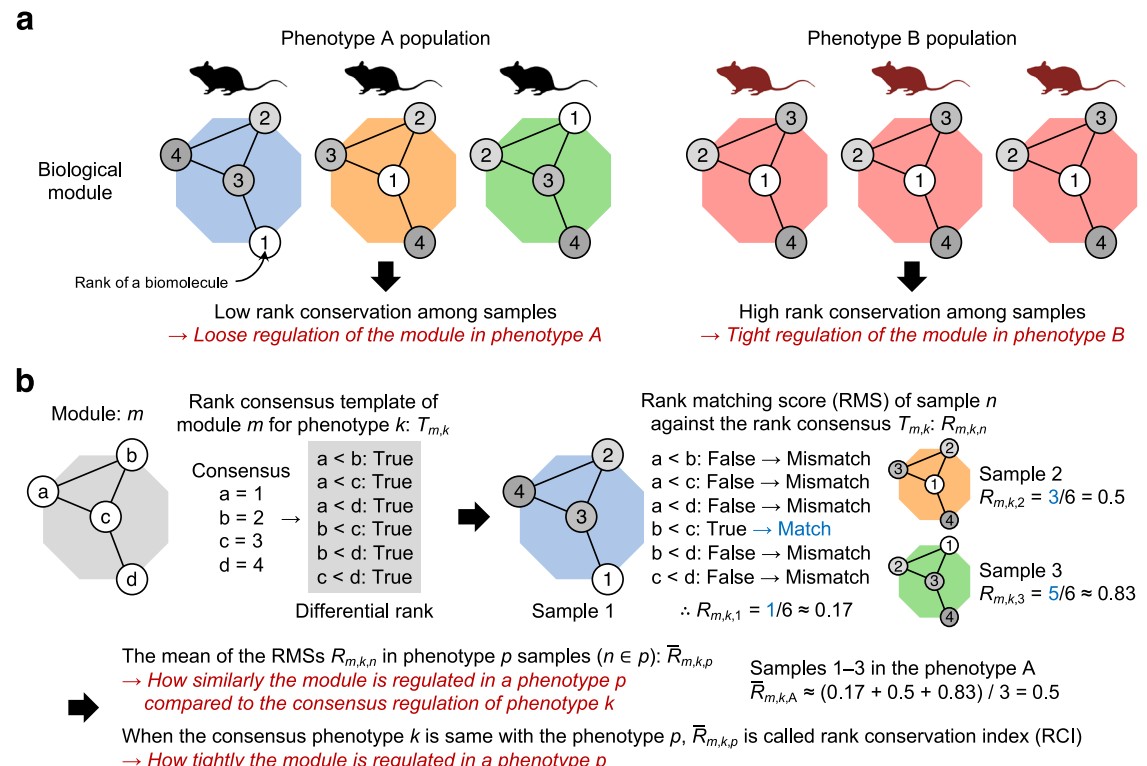

**Fig. 1 Schematic representation of Differential Rank Conservation. a** The concept of rank conservation. A module of a biological system can be represented as a molecular network where nodes and edges indicate biomolecules (e.g., gene transcripts, proteins, and metabolites) and relationships (e.g., physical interactions, chemical reactions), respectively. For each sample, ranks of biomolecules can be obtained from experimental data by ordering the values of interest (e.g., abundances, levels of specific post-translational modification) between the biomolecules within a module. Low and high rank conservation among the samples within a phenotype population suggest loose and tight regulation of the module in the population, respectively. **b** Overview of Differential Rank Conservation (DIRAC) algorithm. DIRAC algorithm utilizes differential ranks of the pairwise rank comparisons within a biological module, and summarizes the rank patterns into sample-level rank matching score (RMS) and population-level rank conservation index (RCI).

The DIRAC metric, rank conservation index (RCI; see Methods)[44], measures population-level consistency in the relative abundance of biomolecules within a module (Fig. 1b); high RCI indicates a strongly shared pattern of behavior (i.e., tight regulation), while low RCI indicates relatively unpatterned behavior (i.e., loose regulation). ACA, 17aE2, and Rapa showed significantly higher RCI median in the examined modules compared to Control (Fig. 2a, Supplementary Fig. 1a), suggesting general tightening of module profiles by each of these lifespan-extending interventions. To identify the modules that were changed (i.e., tightened or loosened) by one or more of the interventions, we assessed the intervention effect on RCI using Analysis of Variance (ANOVA) for each of the 164 modules. There were 23 significantly changed modules (Table 1), based on the false discovery rate (FDR)-adjusted $P < 0.05$ under the conservative assumption that the assessed modules were independent of each other (see Methods; cf. 51 modules exhibited nominal $P < 0.05$). Among these 23 changed modules, the post hoc RCI comparisons between Control and each intervention group revealed that 18, 21, and 22 modules were significantly tightened by ACA, 17aE2, and Rapa, respectively (Fig. 2b, c; Supplementary Fig. 1b), while no module was loosened significantly. 15 modules were significantly tightened under all the three interventions (Fig. 2c), which were functionally related to fatty acid β-oxidation (GO:0006635, GO:0006637, GO:0031998), protein transport to peroxisomes (GO:0006625, GO:0016558), tricarboxylic acid (TCA) cycle (GO:0006102, GO:0006103, GO:0042866), immune response (GO:0098761, GO:0140374), stress response (GO:0006749, GO:0017144, GO:0034063), and translation (GO:0002181, GO:1902416) (Fig. 2d; Supplementary Fig. 2a, c, e, g; Table 1).

Given that the primary modes of action are different among the studied drugs, this result suggests that systems-level regulation for these biological processes may be a general mechanism for lifespan extension.

However, while high RCI reflects a shared pattern of relative abundances within a phenotype population, a tightly regulated module may still exhibit different relative abundance patterns under different phenotypes. The sample-level DIRAC metric, rank matching score (RMS; see Methods)[44], measures the similarity of each sample to the consensus pattern of a certain phenotype (Fig. 1b), and allows us to compare relative abundance patterns between phenotypes by fixing the consensus phenotype (instead of using the sample's own phenotype as the consensus phenotype). For instance, in pyruvate biosynthetic process (GO:0042866) where significantly higher RCI against Control was observed in all the three interventions (Supplementary Fig. 2a), 17aE2 and ACA showed higher and lower mean of RMSs, respectively, than Control under the Rapa rank consensus (Supplementary Fig. 2b), suggesting that 17aE2 changed this module similarly to Rapa while ACA did dissimilarly. Moreover, the RMS mean of 17aE2 was comparable to the RMS mean of Rapa under the Rapa rank consensus (i.e., Rapa's RCI) (Supplementary Fig. 2b). These DIRAC patterns suggest two modes of tightening for this module: one under ACA and the other under 17aE2 and Rapa. Thus, using RMS under each group's rank consensus, we explored the similarity in module tightening across the interventions. Among the 18, 21, and 22 significantly tightened modules by ACA, 17aE2, and Rapa, five, five, and six modules were similarly changed by the other two interventions, respectively (Fig. 2c). In particular, five modules (GO:0006625,

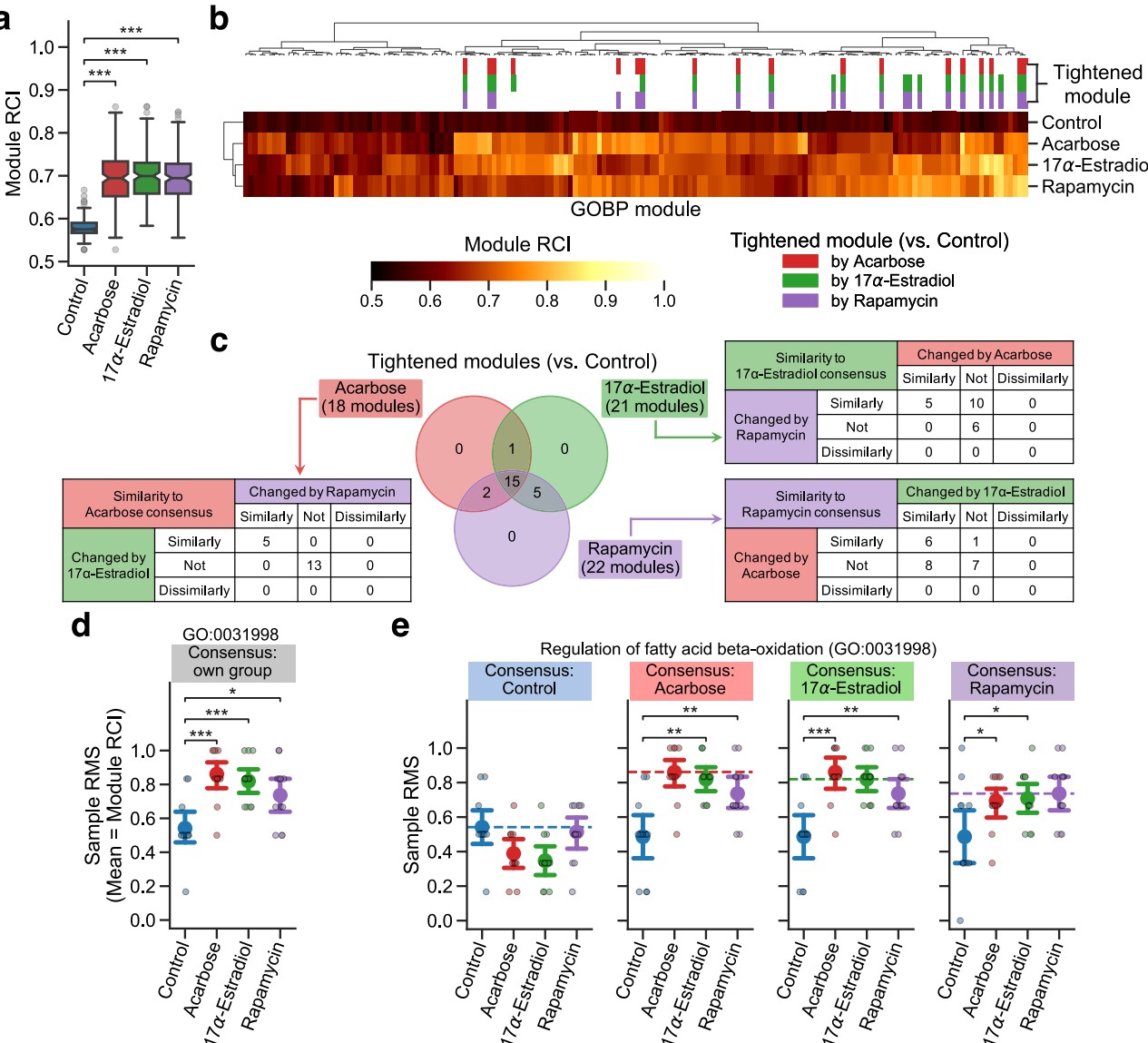

**Fig. 2 Lifespan-extending interventions increased proteomic profile conservation in a priori modules. a–e** Differential Rank Conservation (DIRAC) analysis of the LC-M001 proteomics data using Gene Ontology Biological Process (GOBP)-defined modules (see Supplementary Data 1 for complete results). **a, b** Overall distribution of module rank conservation index (RCI). Data in **a**: median (center line), 95% confidence interval (CI) around median (notch), [$Q_1$, $Q_3$] (box limits), [$x_{min}$, $x_{max}$] (whiskers), where $Q_1$ and $Q_3$ are the 1st and 3rd quartile values, and $x_{min}$ and $x_{max}$ are the minimum and maximum values in [$Q_1 - 1.5 \times IQR$, $Q_3 + 1.5 \times IQR$] (IQR: the interquartile range, $Q_3 - Q_1$), respectively; $n = 164$ modules. ***$P < 0.001$ by two-sided Mann–Whitney $U$-tests after the Benjamini–Hochberg adjustment across three comparisons. The top color columns in **b** highlight the modules that exhibited (1) the significant intervention effect on module RCI (Analysis of Variance (ANOVA) after the Benjamini–Hochberg adjustment across 164 modules) and (2) significantly higher RCI in intervention group than control group (i.e., tightened module; the post hoc two-sided Welch's $t$-tests after the Benjamini–Hochberg adjustment across three comparisons). **c** Venn diagram of the significantly tightened modules by each intervention. For each set of the tightened modules, contingency table indicates the number of modules for which the other intervention groups exhibited significantly higher or lower mean of rank matching scores (RMSs) under the rank consensus than control group (i.e., similarly or dissimilarly changed module to the consensus group, respectively; two-sided Welch's $t$-tests after the Benjamini–Hochberg adjustment across six (two comparisons × three rank consensus) comparisons). **d, e** Sample RMS distributions for an example of the tightened modules (GO:0031998, regulation of fatty acid β-oxidation). Dashed line in **e** indicates the mean of RMSs for the rank consensus group (i.e., RCI). Data: the mean (dot) with 95% CI (bar); $n = 12$ mice. *$P < 0.05$, **$P < 0.01$, ***$P < 0.001$ by two-sided Welch's $t$-tests after the Benjamini–Hochberg adjustment across three (**d**) or six (two comparisons × three rank consensus; **e**) comparisons.

GO:0016558, GO:0031998, GO:0034063, GO:0140374) that were consistently tightened across all the three interventions (Fig. 2c, d; Supplementary Fig. 2c, e, g) exhibited significantly higher mean of RMSs in the intervention groups compared to Control under all the other intervention group's rank consensus (e.g., 17aE2 and Rapa showed significantly higher RMS mean than Control under the ACA rank consensus; Fig. 2e; Supplementary Fig. 2d, f, h; Table 1), suggesting that these biological processes were similarly tightened

by the mechanistically distinct lifespan-extending interventions, and thus can be a general mechanism contributing to longevity.

Taken together, these results suggest that lifespan-extending interventions generally tightened the proteomic profiles of the examined modules and that the tightened protein expression profiles were similar among the different drugs especially in the modules related to fatty acid β-oxidation, peroxisome transport, innate immune response, and stress granule assembly.

**Table 1 Tightened proteomic modules by lifespan-extending interventions.**

| Functional category | Module ID | Module name | ANOVA | Acarbose | 17α-estradiol | Rapamycin | Similarity |
|---|---|---|---|---|---|---|---|
| Fatty acid β-oxidation | GO:0031998 | regulation of fatty acid beta-oxidation | 0.0042 | *** | *** | * | ◎ |
| | GO:0006635 | fatty acid beta-oxidation | 0.0061 | *** | *** | ** | ○ |
| | GO:0006637 | acyl-CoA metabolic process | 0.0245 | ** | *** | ** | |
| | GO:0033539 | fatty acid beta-oxidation using acyl-CoA dehydrogenase | 0.0296 | ** | | ** | |
| | GO:0000038 | very long-chain fatty acid metabolic process | 0.0296 | | * | ** | ○ |
| Protein transport to peroxisome | GO:0006625 | protein targeting to peroxisome | 0.0042 | * | ** | *** | ◎ |
| | GO:0016558 | protein import into peroxisome matrix | 0.0042 | * | ** | *** | ◎ |
| Tricarboxylic acid (TCA) cycle | GO:0006102 | isocitrate metabolic process | 0.0042 | *** | ** | * | |
| | GO:0006103 | 2-oxoglutarate metabolic process | 0.0123 | *** | ** | *** | |
| | GO:0042866 | pyruvate biosynthetic process | 0.0123 | *** | * | * | ○ |
| Immune response | GO:0075522 | IRES-dependent viral translational initiation | 0.0123 | | *** | *** | |
| | GO:0098761 | cellular response to interleukin-7 | 0.0180 | ** | ** | ** | |
| | GO:0140374 | antiviral innate immune response | 0.0292 | * | ** | * | ◎ |
| | GO:0044794 | positive regulation by host of viral process | 0.0370 | | ** | ** | |
| Stress response | GO:0017144 | drug metabolic process | 0.0008 | *** | *** | ** | |
| | GO:0043653 | mitochondrial fragmentation involved in apoptotic process | 0.0042 | | *** | *** | ○ |
| | GO:0006749 | glutathione metabolic process | 0.0042 | *** | *** | ** | |
| | GO:0034063 | stress granule assembly | 0.0123 | ** | *** | ** | ◎ |
| Translation | GO:0002181 | cytoplasmic translation | 0.0168 | * | ** | *** | ○ |
| | GO:0000028 | ribosomal small subunit assembly | 0.0308 | ** | | * | |
| | GO:1902416 | positive regulation of mRNA binding | 0.0324 | * | ** | ** | |
| Miscellaneous | GO:0008210 | estrogen metabolic process | 0.0099 | | ** | *** | ○ |
| | GO:0030042 | actin filament depolymerization | 0.0481 | * | ** | | |

Listed are the 23 Gene Ontology Biological Process (GOBP)-defined proteomic modules that were significantly tightened by one or more of the interventions. ANOVA: P-value of the intervention effect on module rank conservation index (RCI), by Analysis of Variance (ANOVA) after the Benjamini–Hochberg adjustment across 164 modules; Acarbose/17α-estradiol/Rapamycin: significance of the RCI comparison between control and each intervention (*P < 0.05, **P < 0.01, ***P < 0.001), by the post hoc two-sided Welch's t-tests after the Benjamini-Hochberg adjustment across three comparisons; Similarity: significant similarity of the tightening patterns across the interventions (◎: similar to all the other two interventions, ○: similar to other one intervention). Note that the functional categories are just for reference purpose and not completely exclusive with each other. See Supplementary Data 1 for complete results.

**Lifespan-extending interventions increased proteomic profile conservation in data-driven modules.** Considering the potential biases in module definitions within GOBP terms, we inferred data-driven modules using an unsupervised clustering approach, WGCNA[46,47]. WGCNA identifies modules of highly interconnected biomolecules, relying on the overall correlation network computed from high-dimensional data. We applied WGCNA to the LC-M001 proteomics data and identified 10 modules, ranging in size from 68 to 835 proteins (Fig. 3a, Supplementary Data 2). Each WGCNA module can be characterized by the module eigengene (i.e., the first principal component of the protein abundance matrix for the module)[47]. To identify the modules that were associated with one or more of the interventions, we assessed the intervention effect on the module eigengene for each of the 10 data-driven modules using ANOVA. Two modules, denoted Blue and Pink, exhibited a significant intervention effect (FDR-adjusted $P = 0.0028$ for each). The post-hoc comparison in the Blue module revealed that 17aE2 and Rapa, but not ACA, showed a significantly higher value of the module eigengene than Control (Fig. 3b), suggesting that the expression profile of the Blue module was changed specifically by 17aE2 and Rapa. Likewise, the post-hoc comparison suggested that only 17aE2 changed the expression profile of the Pink module (Supplementary Fig. 3a).

WGCNA transforms the correlation matrix to fit a scale-free network topology, where the majority of nodes share relatively few edges with other nodes, while the central nodes that have high intramodular connectivity (called hub nodes) frequently take essential functions in the system[51]. To investigate how 17aE2 and Rapa changed the structure of the Blue and Pink modules, we assessed the relationship between the intervention effect on each module protein and their respective intramodular connectivity (see Methods). The intervention effect on each protein showed significant positive correlation with intramodular connectivity

(Blue: Spearman's $\rho = 0.58$, $P = 3.9 \times 10^{-73}$, Fig. 3c; Pink: Spearman's $\rho = 0.59$, $P = 6.5 \times 10^{-18}$, Supplementary Fig. 3b), suggesting that intramodular hub proteins were more strongly affected by the interventions than less connected proteins. Hence, we further performed enrichment analysis on the hub proteins using GOBP terms (Supplementary Data 3), and found that metabolic processes related to TCA cycle were significantly enriched in the top 10% hub proteins of the Blue module (FDR-adjusted $P = 0.002–0.009$; Fig. 3d). Actually, 23 of the top 30 hub proteins in the Blue module were mitochondrial proteins, involved in TCA cycle, oxidative phosphorylation, and fatty acid β-oxidation (Fig. 3e). Interestingly, prohibitin 1 (PHB1) and PHB2, two of the top five hub proteins, form the mitochondrial PHB complex, which is known to regulate fatty acid oxidation and assembly of mitochondrial respiratory complexes[52,53], as well as to affect lifespan in C. elegans[54]. Likewise, the enrichment analysis revealed that translation processes were significantly enriched in the top 10% hub proteins of the Pink module (FDR-adjusted $P \le 10^{-7}$; Supplementary Fig. 3c), and 21 of the top 30 hub proteins in the Pink module were actually ribosomal proteins (Supplementary Fig. 3d). Collectively, our results from WGCNA revealed coordinated changes in the expression profiles of mitochondrial and ribosomal proteins that were limited to 17aE2 or Rapa among the three studied drugs.

Next, we re-analyzed the LC-M001 proteomics data using the DIRAC method with six of the 10 WGCNA-identified modules (see Methods; Fig. 3a, Supplementary Data 4). ACA, 17aE2, and Rapa showed significantly higher RCI median in the examined modules than Control (Fig. 3f), suggesting general tightening of module profiles by each of these lifespan-extending interventions, consistent with the initial DIRAC result based on GOBP terms (Fig. 2a). Additionally, all six WGCNA modules exhibited significant intervention effects on RCI in ANOVA (FDR-adjusted $P < 10^{-5}$) and significantly higher RCIs for intervention groups

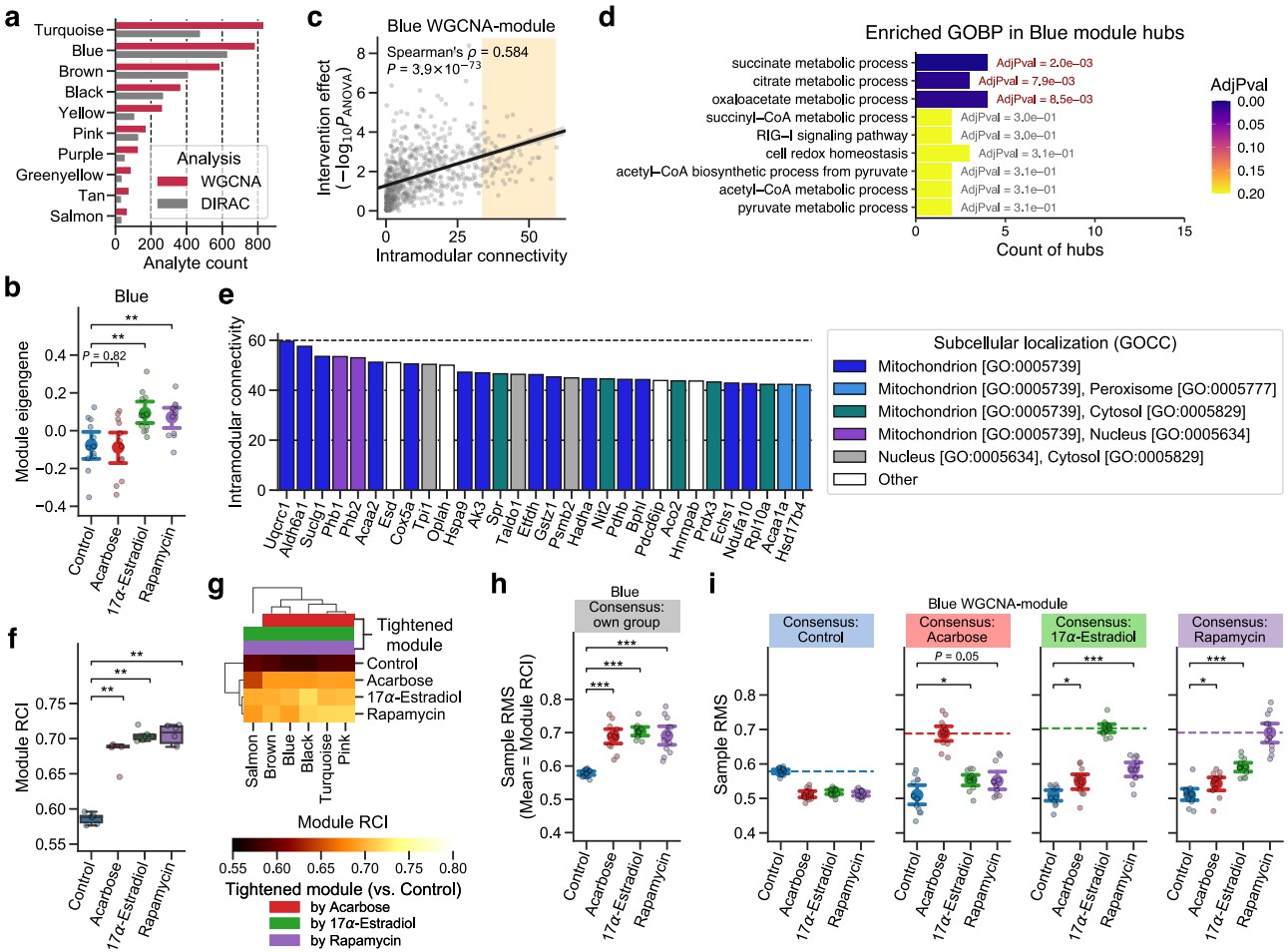

**Fig. 3 Lifespan-extending interventions increased proteomic profile conservation in data-driven modules. a–e** Weighted Gene Co-expression Network Analysis (WGCNA) of the LC-M001 proteomics data (see Supplementary Data 2 and 3 for complete results). **a** The number of proteins in each WGCNA-identified module. WGCNA: proteins used in WGCNA, DIRAC: proteins retained after the processing for Differential Rank Conservation (DIRAC) analysis **f–i**. **b** Distributions of sample's module eigengene values for the Blue module. Data: the mean (dot) with 95% confidence interval (CI) (bar); $n = 12$ mice. **$P < 0.01$ by two-sided Welch's $t$-tests after the Benjamini–Hochberg adjustment across three comparisons. **c** Relationship between the intervention effect on each protein and their respective intramodular connectivity in the Blue module. The $P$-value of y-axis corresponds to the main effect of intervention on each protein level by Analysis of Variance (ANOVA). The line is the ordinary least squares (OLS) linear regression line with 95% CI, and the orange-colored background reflects the range of top 10% hub proteins (79 proteins). $n = 787$ proteins. **d** Enriched Gene Ontology Biological Process (GOBP) terms in the top 10% hub proteins of the Blue module. Significance was assessed using overrepresentation tests after the Benjamini–Hochberg adjustment across 81 terms. Only the GOBP terms that exhibited nominal $P < 0.05$ are presented. AdjPval: adjusted $P$-value from the overrepresentation test. **e** Top 30 hub proteins of the Blue module. GOCC: GO Cellular Component. **f–i** DIRAC analysis of the LC-M001 proteomics data using WGCNA-identified modules (see Supplementary Data 4 for complete results). **f, g** Overall distribution of module rank conservation index (RCI). Data in **f**: median (center line), $[Q_1, Q_3]$ (box limits), $[x_{min}, x_{max}]$ (whiskers), where $Q_1$ and $Q_3$ are the 1st and 3rd quartile values, and $x_{min}$ and $x_{max}$ are the minimum and maximum values in $[Q_1 - 1.5 \times IQR, Q_3 + 1.5 \times IQR]$ (IQR: the interquartile range, $Q_3 - Q_1$), respectively; $n = 6$ modules. **$P < 0.01$ by two-sided Mann–Whitney $U$-tests after the Benjamini–Hochberg adjustment across three comparisons. The top color columns in **g** highlight the modules that exhibited (1) the significant intervention effect on module RCI (ANOVA after with the Benjamini–Hochberg adjustment across six modules) and (2) significantly higher RCI in intervention group than control group (i.e., tightened module; the post hoc two-sided Welch's $t$-tests after the Benjamini–Hochberg adjustment across three comparisons). **h, i** Sample rank matching score (RMS) distributions for the Blue module. Dashed line in **i** indicates the mean of RMSs for the rank consensus group (i.e., RCI). Data: the mean (dot) with 95% CI (bar); $n = 12$ mice. *$P < 0.05$, ***$P < 0.001$ by two-sided Welch's $t$-tests after the Benjamini–Hochberg adjustment across three (**h**) or six (two comparisons × three rank consensus; **i**) comparisons.

against Control in the post hoc RCI comparisons (Fig. 3g, h; Supplementary Fig. 3e), suggesting that all the WGCNA modules were tightened by ACA, 17aE2, or Rapa. Moreover, the mitochondrial Blue module exhibited higher mean of RMSs in intervention groups compared to Control under all the other intervention group's rank consensus (e.g., ACA and 17aE2 showed significantly higher RMS mean than Control under the Rapa rank consensus; Fig. 3i), suggesting that the tightening patterns were similar among the three drugs. At the

same time, the Rapa's RMS mean was more similar to the 17aE2's RMS mean than to the ACA's RMS mean (e.g., 17aE2 showed higher RMS mean than ACA under the Rapa rank consensus) in the Blue module (Fig. 3i), implying a difference in the tightening pattern between ACA vs. 17aE2 and Rapa, in line with their effects on module expression profiles (Fig. 3b). Note that, if the tightening pattern of ACA was completely dissimilar to those of 17aE2 and Rapa, ACA would have shown lower RMS mean than Control under the 17aE2 or

Rapa rank consensus (cf. compare the patterns between Fig. 3i and Supplementary Fig. 2b). Likewise, the ribosomal Pink module exhibited similar tightening patterns among the three drugs (Supplementary Fig. 3f).

These findings suggest that the Blue and Pink modules were tightened across all the three interventions, while also exhibiting intervention-specific effects on protein expression profiles related to mitochondrial energy metabolism and ribosomal translation. Taken together, our results indicate that the tightening of module profiles was a general signature of the lifespan-extending interventions within the measured proteomic space.

**Lifespan-extending interventions increased the conservation of proteomic modules through transcriptional or post-transcriptional alteration.** To further investigate the tightening effects of lifespan-extending interventions on proteomic modules, we applied DIRAC analysis to a mouse liver transcriptomic dataset from a previous study of lifespan-extending interventions[45] (referred as M001-related transcriptomics), whose experimental design resembled the LC-M001 experiment (e.g., the same animal colony, the same genetically heterogeneous mouse stock for pharmacological intervention groups, and similar protocol to the NIA ITP protocol) and contained ACA, 17aE2, and Rapa treatments as lifespan-extending interventions. In this M001-related experiment, 78 mice were prepared for either control or one of lifespan-extending interventions (two genetically modified models, two nutritional interventions, and four pharmacological interventions) and euthanized at young adult ages depending on the intervention type ($n = 3$–12 mice per intervention group). Following the similar considerations to those in the proteomics analyses, we regressed out the potential effects of sex and age (see Methods), pooled samples for each intervention (see Discussion), and analyzed the three lifespan-extending interventions (ACA, Rapa, and CR) and their corresponding control (Control) throughout the current study ($n = 12$ (6 female and 6 male) mice per group).

Using 3912 a priori modules that were defined by mapping the measured transcripts to GOBP annotations (see Methods; Supplementary Data 5), we found that ACA, Rapa, and CR showed significantly higher RCI median in the examined modules than Control (Fig. 4a). This result suggests the general tightening of module profiles within the measured transcriptomic space, as well as within the measured proteomic space (Figs. 2a and 3f). We next assessed the intervention effect on RCI using ANOVA for each of the 3912 modules, and identified 2516 modules that were significantly changed by one or more of the interventions (FDR-adjusted $P < 0.05$; see Methods; cf. 2691 modules exhibited nominal $P < 0.05$). Among these 2516 changed modules, the post hoc RCI comparisons between Control and each intervention group revealed that 1662, 1424, and 2454 modules were significantly tightened by ACA, Rapa, and CR, respectively (Supplementary Fig. 4a, b), while no module was loosened significantly. Subsequently, using RMS under each group's rank consensus, we explored the similarity of module tightening across the interventions. Among the 1662, 1424, and 2454 significantly tightened modules by ACA, Rapa, and CR, 139, 81, and 54 modules were similarly changed by the other two interventions, respectively (Supplementary Fig. 4b). For instance, ubiquitin-dependent protein catabolic process (GO:0006511), a consistently tightened module across the interventions (Supplementary Fig. 4c), exhibited significantly higher mean of RMSs in intervention groups compared to Control under all the other intervention group's rank consensus (e.g., Rapa and CR showed significantly higher RMS mean than Control under the ACA rank consensus; Supplementary Fig. 4d), suggesting that this module was similarly tightened in transcripts by the mechanistically distinct lifespan-extending interventions.

To directly compare the DIRAC results between the LC-M001 proteomics and M001-related transcriptomics datasets, we focused on the two interventions (ACA and Rapa) and the 156 GOBP modules that were used in both omics results (Supplementary Data 6), and re-assessed the intervention effect on RCI using ANOVA for each of the 156 modules and each omics. There were 33 modules (22 in proteins and 16 in transcripts) that were significantly changed by one or more of the interventions (FDR-adjusted $P < 0.05$; see Methods). Among these 33 changed modules, the post hoc RCI comparisons between Control and each intervention group revealed that 25, 22, 27, and 18 modules were significantly tightened by ACA in proteins, ACA in transcripts, Rapa in proteins, and Rapa in transcripts, respectively (Fig. 4b). In particular, the modules that were significantly tightened by ACA and Rapa in both proteins and transcripts were nine modules, including processes related to fatty acid β-oxidation (GO:0006635, GO:0006734, GO:0015909), peroxisome transport (GO:0016558), immune response (GO:0098761), or stress response (GO:0006749) (Fig. 4b, c). This result suggests that these modules were tightened by the lifespan-extending interventions via transcriptional regulation with concordant changes of proteomic profiles. At the same time, we also observed seven modules which were tightened specifically in proteins (Fig. 4b). For instance, pyruvate biosynthetic process (GO:0042866) exhibited significantly higher RCI across interventions compared to Control specifically in proteins (Fig. 4d), suggesting that this module was tightened by ACA and Rapa in the proteomic profile but not in the transcriptomic profile. This inconsistency between proteins and transcripts may reflect post-transcriptional regulatory mechanisms that can affect protein profiles beyond transcriptional changes. For instance, since the abundance of a protein is determined by both its synthesis and degradation rates, a difference in proteostasis, whose loss is proposed as a characteristic of aging[1,55,56], can lead to the change in protein abundance without a change in transcript abundance.

Altogether, these findings suggest that the tightening of module profiles was a general signature of the lifespan-extending interventions also at the transcript level, while some modules were tightened uniquely at the protein level potentially through post-transcriptional alteration.

**Lifespan-extending interventions increased proteomic profile conservation of specific modules potentially through cap-independent translation.** CIT of specific mRNAs[48] can be a possible post-transcriptional mechanism to explain the inconsistency in module tightness between proteins and transcripts. In contrast to standard cap-dependent translation, CIT does not require the interaction of the eukaryotic initiation factor 4E (eIF4E) complex with the 5′ cap of mRNA; $N^6$-methyladenosine (m$^6$A) modification in the 5′ untranslated regions of mRNA can trigger the recruitment of specific initiation and elongation factors, followed by the selective translation of m$^6$A-tagged mRNAs. Previous studies have shown the upregulated translation of a subset of mRNAs via CIT in long-lived endocrine mutant mice[57] and similar increases of CIT in mice treated with ACA, 17aE2, or Rapa[58]. We therefore assessed if CIT could explain the difference in the module tightness between proteins and transcripts, by utilizing DIRAC analysis on another liver proteomic dataset which was generated through a mouse CIT experiment (denoted LC-M004 proteomics). In this LC-M004 experiment, 16 young adult mice were treated with either solvent (Control-2) or 4EGI-1, a synthetic small compound which inhibits the eIF4E–eIF4G interaction and thereby blocks cap-dependent translation and enhances CIT[59] ($n = 8$ (4 female and 4 male) mice per group). To directly compare the DIRAC results across the LC-M001 and LC-M004 proteomics datasets, we focused on 153 GOBP modules for

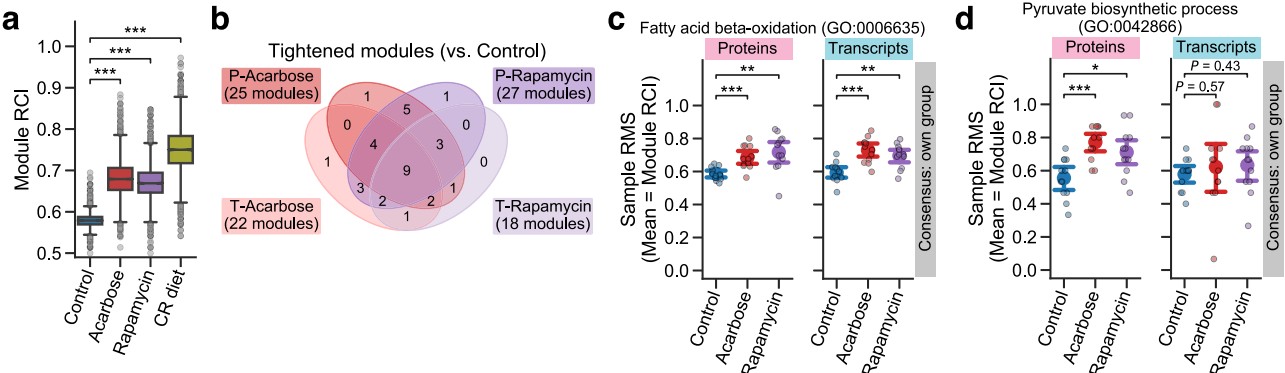

**Fig. 4 Lifespan-extending interventions increased the conservation of proteomic modules through transcriptional or post-transcriptional alteration.**
**a** Differential Rank Conservation (DIRAC) analysis of the M001-related transcriptomics data using Gene Ontology Biological Process (GOBP)-defined modules (see Supplementary Data 5 for complete results). Presented is the overall distribution of module rank conservation index (RCI). CR: calorie restriction. Data: median (center line), 95% confidence interval (CI) around median (notch), $[Q_1, Q_3]$ (box limits), $[x_{min}, x_{max}]$ (whiskers), where $Q_1$ and $Q_3$ are the 1st and 3rd quartile values, and $x_{min}$ and $x_{max}$ are the minimum and maximum values in $[Q_1 - 1.5 \times IQR, Q_3 + 1.5 \times IQR]$ (IQR: the interquartile range, $Q_3 - Q_1$), respectively; $n = 3912$ modules. ***$P < 0.001$ by two-sided Mann–Whitney $U$-tests after the Benjamini–Hochberg adjustment across three comparisons. **b–d** DIRAC comparison analysis between the LC-M001 proteomics and M001-related transcriptomics data using GOBP-defined modules (see Supplementary Data 6 for complete results). **b** Venn diagram of the modules that exhibited (1) the significant intervention effect on module RCI (Analysis of Variance (ANOVA) after the Benjamini–Hochberg adjustment across 312 (156 modules × two omics datasets) tests) and (2) significantly higher RCI in intervention group than control group (i.e., tightened module; the post hoc two-sided Welch's $t$-tests after the Benjamini–Hochberg adjustment across four (two comparisons × two omics datasets) comparisons). P: proteomics, T: transcriptomics. **c, d** Sample rank matching score (RMS) distributions for an example of the tightened modules in both proteins and transcripts (**c**; GO:0006635, fatty acid β-oxidation) or specifically in proteins (**d**; GO:0042866, pyruvate biosynthetic process). Data: the mean (dot) with 95% CI (bar); $n = 12$ mice. *$P < 0.05$, **$P < 0.01$, ***$P < 0.001$ by two-sided Welch's $t$-tests after the Benjamini–Hochberg adjustment across four (two comparisons × two omics datasets) comparisons.

this analysis, which were mapped to the measured proteins in both datasets (see Methods; Supplementary Data 7). In addition to the elevated module tightness in ACA, 17aE2, and Rapa against their corresponding control (Control-1), 4EGI-1 showed significantly higher RCI median in the examined modules compared to Control-2 (Fig. 5a), implying general tightening of module profiles by the CIT enhancement (in line with the expectation from 4EGI-1's mode of action).

To reveal the similarity of module profiles between lifespan-extending interventions and 4EGI-1, we calculated RMSs under the rank consensus of Control-2 and 4EGI-1 for the LC-M001 groups (Control-1, ACA, 17aE2, and Rapa), and assessed the intervention effect on the RMS mean using ANOVA for each of the 153 modules and each rank consensus. There were five and nine modules that were significantly changed by one or more of the interventions under the Control-2 and 4EGI-1 rank consensus, respectively (FDR-adjusted $P < 0.05$; see Methods; cf. 40 and 34 modules exhibited nominal $P < 0.05$, respectively). Among these nine changed modules under the 4EGI-1 rank consensus, the post hoc comparisons for the RMS mean between Control-1 and each intervention group revealed that one, zero, and one module were changed dissimilarly to the 4EGI-1 consensus by ACA, 17aE2, and Rapa, respectively (Fig. 5b). For instance, in positive regulation by host of viral process (GO:0044794), Rapa showed significantly lower mean of RMSs than Control-1 under the 4EGI-1 consensus (Fig. 5c). Given that this module was significantly tightened by Rapa in proteins but not in transcripts (Fig. 5d), it suggests that this virus-related process may be tightened in proteins via post-transcriptional regulation other than CIT. In contrast, the post hoc RMS mean comparisons for the nine changed modules also revealed that one, three, and two modules were changed similarly to the 4EGI-1 consensus by ACA, 17aE2, and Rapa, respectively (Fig. 5e). Remarkably, in isocitrate metabolic process (GO:0006102), ACA and 17aE2 showed significantly higher mean of RMSs than Control-1 under the 4EGI-1 rank consensus (Fig. 5f), while ACA

showed significantly higher RCI than control in proteins but not in transcripts (Fig. 5g), suggesting that ACA may tighten proteomic profiles of this TCA cycle-related process via CIT. Collectively, these findings suggest that not all but some specific modules may be tightened by lifespan-extending interventions through augmented CIT.

**Lifespan-extending interventions shifted reaction fluxes in fatty acid metabolism.** Finally, as a complementary approach to the findings from DIRAC analysis and WGCNA, we performed in silico analysis using the mouse GEM[49] to investigate metabolic shifts associated with lifespan-extending interventions. GEM is a mathematical framework that leverages a curated catalog of biochemical reactions within a system (e.g., single cell, tissue, organ), including metabolites, catalytic enzyme-encoding genes, and their stoichiometry[50]. Using optimization techniques with large-scale experimental data (e.g., transcriptomics), the solved stoichiometric coefficients of each reaction allow flux prediction for metabolic reactions in the system at equilibrium[60]. Hence, GEM has been used to investigate metabolic changes in various systems and specific contexts (e.g., human cancers)[61]. Since the detected proteins in the LC-M001 proteomics did not sufficiently cover the metabolic proteins included in the mouse GEM, we utilized the aforementioned M001-related transcriptomics data[45] where the potential effects of sex and age were regressed out (Control, ACA, Rapa, and CR; $n = 12$ (6 female and 6 male) mice per group). By integrating it with the mouse generic GEM[49] for each of the 48 samples, we generated 48 context-specific metabolic networks (i.e., GEMs constrained by each sample condition), and subsequently predicted flux values of the metabolic reactions for each context-specific GEM (see Methods). As a result, the flux values were successfully predicted for 7834 reactions among the 10,612 reactions of the generic GEM, and only the 3736 functional reactions were assessed (see Methods; Supplementary Data 8).

In the GEM system, the reactions exhibit a high degree of interconnectivity, making it overly conservative to account for

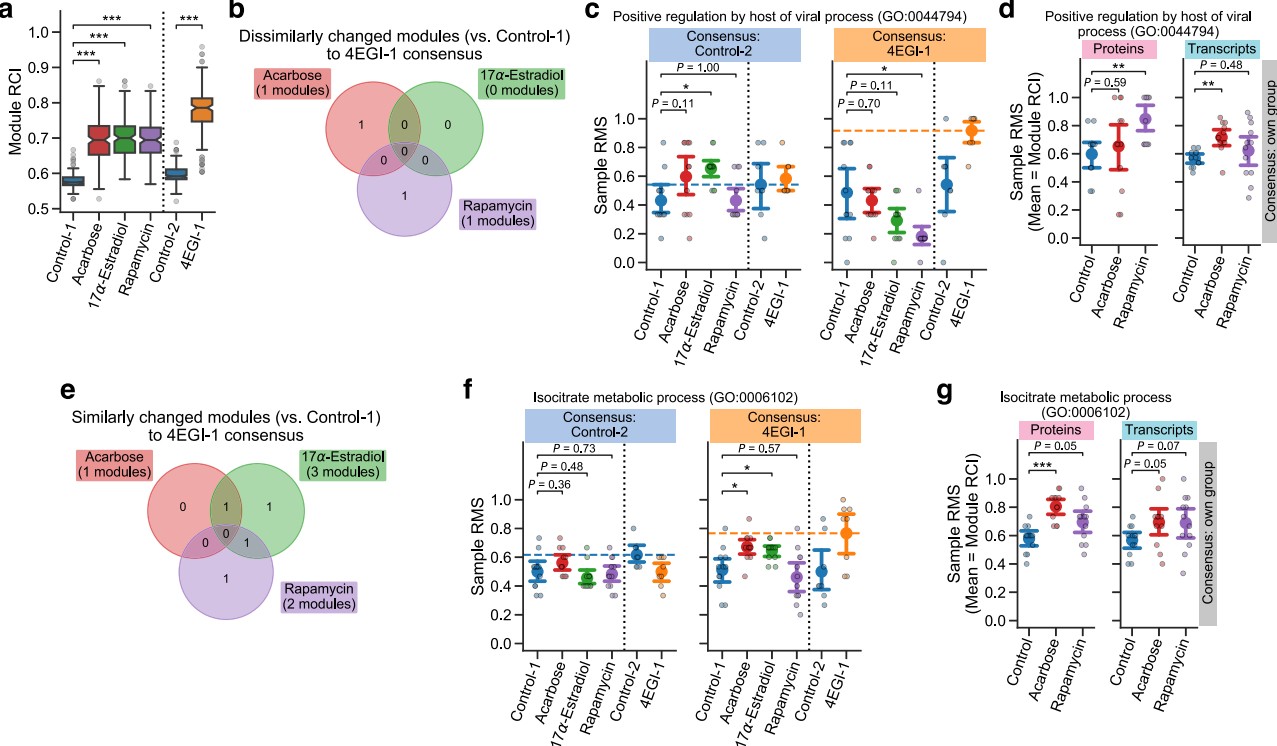

**Fig. 5 Lifespan-extending interventions increased proteomic profile conservation of specific modules potentially through cap-independent translation.**
**a–g** Differential Rank Conservation (DIRAC) analysis of the LC-M001 and LC-M004 proteomics data (**a–c**, **e**, **f**) or DIRAC comparison analysis between the LC-M001 proteomics and M001-related transcriptomics data (**d**, **g**) using Gene Ontology Biological Process (GOBP)-defined modules (see Supplementary Data 7 (**a–c**, **e**, **f**) or 6 (**d**, **g**) for complete results, respectively). Control-1: control for Acarbose, 17α-Estradiol, and Rapamycin; Control-2: control for 4EGI-1. **a** Overall distribution of module rank conservation index (RCI). Data: median (center line), 95% confidence interval (CI) around median (notch), $[Q_1, Q_3]$ (box limits), $[x_{min}, x_{max}]$ (whiskers), where $Q_1$ and $Q_3$ are the 1st and 3rd quartile values, and $x_{min}$ and $x_{max}$ are the minimum and maximum values in $[Q_1 - 1.5 \times IQR, Q_3 + 1.5 \times IQR]$ (IQR: the interquartile range, $Q_3 - Q_1$), respectively; $n = 153$ modules. \*\*\*$P < 0.001$ by two-sided Mann–Whitney $U$-tests after the Benjamini–Hochberg adjustment across four comparisons. **b**, **e** Venn diagram of the modules that exhibited (1) the significant intervention effect on module mean of RMSs under 4EGI-1 rank consensus (Analysis of Variance (ANOVA) after the Benjamini–Hochberg adjustment across 306 (153 modules × two rank consensus) tests) and (2) significantly lower (**b**) or higher (**e**) mean of RMSs in intervention group than control group (i.e., dissimilarly (**b**) or similarly (**e**) changed module to the 4EGI-1 group; the post hoc two-sided Welch's $t$-tests after the Benjamini–Hochberg adjustment across six (three comparisons × two rank consensus) comparisons. **c**, **d**, **f**, **g** Sample rank matching score (RMS) distributions for an example of the dissimilarly tightened modules (**c**, **d**; GO:0044794, positive regulation by host of viral process) or the similarly tightened modules (**f**, **g**; GO:0006102, isocitrate metabolic process). Dashed line in **c** and **f** indicates the mean of RMSs for the rank consensus group (i.e., RCI). Data: the mean (dot) with 95% CI (bar); $n = 8$ (Control-2, 4EGI-1), 12 (the others) mice. \*$P < 0.05$, \*\*$P < 0.01$, \*\*\*$P < 0.001$ by two-sided Welch's $t$-tests after the Benjamini–Hochberg adjustment across six (three comparisons × two rank consensus; **c**, **f**) or four (two comparisons × two omics datasets; **d**, **g**) comparisons.

multiple testing by assuming independence among the reactions. In addition, the number of mapped reactions to a subsystem (i.e., analogous to functional pathways within the GEM system) is different between subsystems[49]. Hence, we assessed which subsystems in GEM were shifted by each intervention as follows. First, we screened for the reactions that were potentially changed by an intervention, using Mann–Whitney $U$-test with nominal $P < 0.05$ (Fig. 6a). Subsequently, we performed enrichment analysis on the potentially changed reactions using GEM subsystem annotations. As a result, six and five subsystems were significantly enriched in the 170 and 38 potentially changed reactions by ACA and CR, respectively (FDR-adjusted $P < 0.05$; Fig. 6b, c), while no potentially changed reaction (and thus no enriched subsystem) was identified for Rapa. In particular, consistently enriched between ACA and CR was the fatty acid oxidation subsystem (Fig. 6b, c). Moreover, when focusing on the central energy metabolism, the reaction chains around pyruvate, acetyl-CoA, and acylcarnitine were significantly changed by ACA with tendency of concordant changes by CR (Fig. 6d). In summary, our in silico analysis identified the metabolic effects of different lifespan-extending interventions (i.e., ACA and CR at

least), and implied that multiple lifespan-extending interventions concordantly shifted fatty acid metabolism at the systems level.

## Discussion

Studies in model organisms, both invertebrate and vertebrate, have shown multiple ways to extend lifespan and delay age-related diseases[3,14–16]. Aging can be slowed, and healthspan can be extended, by mutation of individual genes, dietary restrictions, or oral administration of compounds. Data are becoming available to determine which of the many cellular and molecular traits modified by each of these interventions are shared across slow-aging models and which are specific. Elucidation of the physiological and cellular mechanisms of effective interventions will provide clues for possible measures to improve human health and may also give useful prognostic information. In this study, we demonstrated the following key findings: (1) mechanistically distinct lifespan-extending interventions generally tightened the systems-level profiles of biological processes; (2) fatty acid metabolism emerged as a common process shifted by multiple lifespan-extending interventions; (3) lifespan-extending interventions achieved the tight proteomic

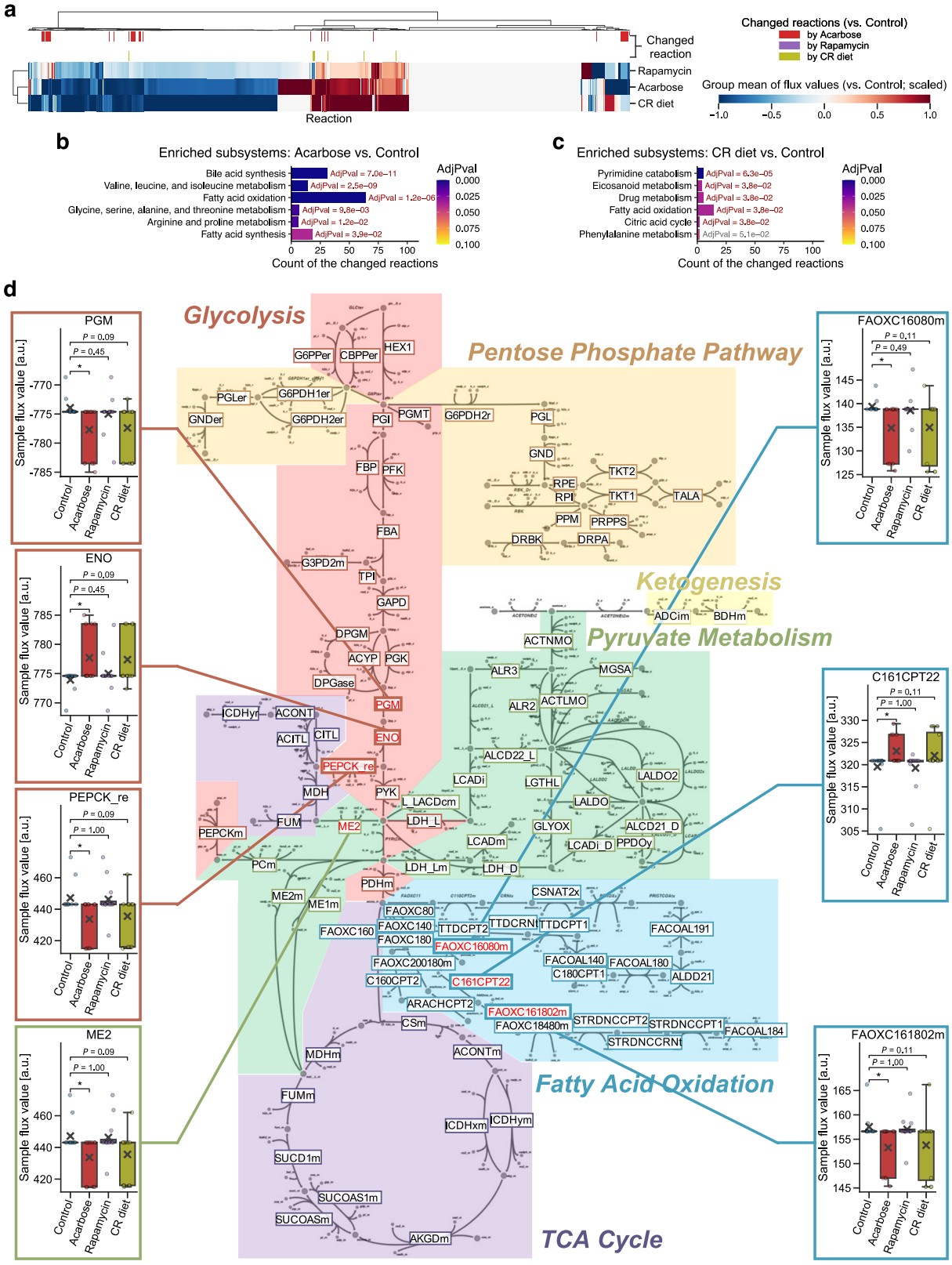

profiles of biological processes through transcriptional or post-transcriptional regulation, potentially including CIT.

By leveraging omic datasets and systems-level approaches jointly, we demonstrated that lifespan-extending interventions modified biological processes and metabolic reactions at the systems level (Figs. 2–6). In particular, DIRAC analyses revealed that the tightening of module profiles was a general signature of the lifespan-extending interventions within the measured proteomic and transcriptomic spaces (Figs. 2a, 3f and 4a). Interestingly, a previous study using DIRAC revealed the general loosening of module profiles in more malignant phenotypes and later stages of cancer progression among various cancer types[44]. Given that there are some similarities between cancer resistance and anti-aging in mechanisms (e.g., DNA repair, telomere maintenance)[55,56,62], aging

**Fig. 6 Lifespan-extending interventions shifted reaction fluxes in fatty acid metabolism. a–d** Integrated analysis of the M001-related transcriptomics data with mouse genome-scale metabolic model (GEM; see Supplementary Data 8 for complete results). CR: calorie restriction. **a** Change in the group mean of flux values for each reaction. The presented group mean value was centered and scaled per reaction (see Methods); i.e., its positive value corresponds to an increase in the mean of flux values compared to the corresponding control group, and vice versa. The top color columns highlight the reactions that exhibited difference in flux value distribution between control and intervention (i.e., potentially changed reaction; nominal $P < 0.05$ by two-sided Mann–Whitney $U$-test). $n = 3736$ reactions. **b, c** Shifted GEM subsystems by acarbose (**b**) or CR diet (**c**). Significance was assessed as the enrichment of GEM subsystem in the potentially changed reactions, using overrepresentation tests with the Benjamini–Hochberg adjustment across 21 (**b**) or 11 (**c**) subsystems. Only the subsystems that exhibited nominal $P < 0.05$ are presented. AdjPval: adjusted $P$-value from the overrepresentation test. **d** Changed reactions within the central energy metabolism. The 102 highlighted reaction IDs are the reactions assessed in this study. Flux value distributions are presented only for the potentially changed reactions within this diagram. Data: mean (cross sign), median (center line), $[Q_1, Q_3]$ (box limits), $[x_{min}, x_{max}]$ (whiskers), where $Q_1$ and $Q_3$ are the 1st and 3rd quartile values, and $x_{min}$ and $x_{max}$ are the minimum and maximum values in $[Q_1 - 1.5 \times IQR, Q_3 + 1.5 \times IQR]$ (IQR: the interquartile range, $Q_3 - Q_1$), respectively; $n = 12$ mice. $^*P < 0.05$ by two-sided Mann–Whitney $U$-test after the Benjamini–Hochberg adjustment across three comparisons.

may be promoted, in part, by loss of tight regulation for pertinent modules, and maintenance of the module tightness may be a key anti-aging strategy. Furthermore, we identified the 23 proteomic modules (Fig. 2c, Table 1, Supplementary Data 1), 2500 transcriptomic modules (Supplementary Fig. 4b, Supplementary Data 5), and 10 GEM subsystems (Fig. 6b, c; Supplementary Data 8) that were affected by one or more of the lifespan-extending interventions. These modules and subsystems included biological processes related to fatty acid β-oxidation, TCA cycle, immune response, stress response, and proteostasis, consistent with well investigated roles in aging[7–10,55,56]. It should be noted that these processes have already been regarded as components of the aging hallmarks[55,56] and the pillars of aging[1]. However, while these hallmarks and pillars were compiled from independent (and, in many cases, hypothesis-driven) studies and are often criticized[5], our findings were deduced in a data-driven manner with an untargeted approach. Therefore, our current study showed the power of systems-level approaches to explore and test hypotheses about the control of aging and longevity in mammals, and provided a translational implication that potential lifespan-extending interventions may be identified and evaluated based on their regulatory effects on these systems.

Fatty acid β-oxidation is the catabolic process of fatty acid breakdown for energy production, with mitochondria and peroxisomes being the major involved organelles[63]. We demonstrated that fatty acid β-oxidation was consistently tightened in both proteins and transcripts across mechanistically distinct lifespan-extending interventions (Figs. 2d and 4c). We also observed that the system transporting proteins into peroxisomes was tightened in proteins consistently across the interventions (Supplementary Fig. 2c), and implied that mitochondrial energy metabolism was affected by 17aE2 and Rapa (Fig. 3b–e). Moreover, we showed that reactions involved in fatty acid metabolism were concordantly shifted in ACA and CR (Fig. 6b–d). All these findings support the conclusion that fatty acid β-oxidation was directed towards tight control in a whole cellular system for longevity. At the same time, this systems-level control of fatty acid β-oxidation may be achieved through different mechanisms by each intervention. For example, although fatty acid β-oxidation (GO:0031998) was similarly tightened by ACA, 17aE2, and Rapa (Fig. 2e), the tightening patterns in a broader fatty acid β-oxidation module (GO:0006635) and essentially connected modules such as pyruvate synthesis (GO:0042866) and fatty acid metabolism (GO:0000038) were similar between 17aE2 and Rapa, but different from ACA (Supplementary Fig. 2b, Table 1). 17aE2 and Rapa, but not ACA, similarly modulated expression patterns of mitochondrial proteins (Fig. 3b). Rapa showed no (significant) flux changes in fatty acid β-oxidation (Fig. 6d). Hence, we hypothesize that different lifespan-extending interventions lead to a similar rerouting of energy metabolism

through fatty acid metabolism, albeit through different mechanisms. Although the findings from DIRAC, WGCNA, and GEM do not indicate the functional consequence for cells (e.g., tight regulation can be either augmentation or attenuation of the pathway), there are multiple reports highlighting the role of fatty acid oxidation in aging and longevity. For instance, AMPK, an essential kinase of the nutrient-sensing signaling pathways in longevity, inhibits fatty acid synthesis and promotes fatty acid oxidation via inhibition of acetyl-CoA carboxylase 1 (ACC1) and ACC2[7,64]. CR increases fatty acid synthesis in adipose tissue but results in enhancing whole-body oxidation[65]. Ketogenic diet specifically upregulates the genes involved in fatty acid oxidation in liver[19]. Overexpression of fatty acid-binding protein (FABP) or dodecenoyl-CoA delta-isomerase (DCI), corresponding to the acceleration of fatty acid β-oxidation, increased lifespan in *D. melanogaster*[66]. Therefore, tight regulation promoting fatty acid β-oxidation could be a common signature among lifespan-extending strategies. The prominence of the nutrient-sensing or energy-producing process from liver-derived datasets might be unsurprising, given that the liver is a major metabolic organ. However, we also observed that lifespan-extending interventions tightened the modules less often associated with liver and metabolism, such as antiviral innate immune response (GO:0140374; Supplementary Fig. 2e), positive regulation by host of viral process (GO:0044794; Fig. 5d), and cellular response to interleukin-7 (GO:0098761; Table 1). In the mid-life human female brain, metabolic and immune systems are shifted by chronological age: glucose metabolism and fatty acid β-oxidation are attenuated and enhanced, respectively, and chronic low-grade innate and adaptive immune responses are enhanced[67]. Hence, the interrelationship between fatty acid metabolism and innate/adaptive inflammation is an interesting area for future investigations.

There are some limitations to this study. We pooled female and male samples due to the sample size limitation. Hence, it is highly possible that we failed to identify sex-dependent changes, especially related to the known sex-dependent effects of ACA and 17aE2 on lifespan extension[23–25]. Additionally, our findings relied on liver datasets, and the systemic regulation of modules across organs and tissues was not addressed. Because this study successfully validated the utility of systems-level approaches and because sex dimorphism and cross-systems regulation remain not fully elucidated in aging and longevity[68,69], we plan to address this point in future studies as datasets containing larger sample sizes and multiple tissues become available.

In summary, this study consolidates the previously fragmented interrelationships among various lifespan-extending interventions into a more coherent framework. It emphasizes the effectiveness of systems-level approaches in identifying and characterizing the biological processes involved in aging and longevity.

## Methods

**Mouse liver proteomic datasets**. Liver samples from mice fed with lifespan-extending drugs were collected as previously described[58]. Briefly, 12 (6 female and 6 male) genetically heterogeneous UM-HET3 mice were prepared for each sample group: control, acarbose (ACA), 17α-estradiol (17aE2), and rapamycin (Rapa). The drugs were treated via daily feeding of the Purina 5LG6 diet with ACA (1000 mg kg$^{-1}$), 17aE2 (14.4 mg kg$^{-1}$), or Rapa (14 mg kg$^{-1}$) starting at 4 months. At 12 months, the mice fasted for 18 h and were euthanized for liver sampling. Excised livers were washed in phosphate-buffered saline (PBS) and snap-frozen for proteomic analysis. All procedures followed the methods recommended by the National Institute on Aging (NIA) Interventions Testing Program (ITP)[22]. Hereinafter, this experiment is called LC-M001. Liver samples from 4EGI-1-treated mice were collected as previously described[57]. Each group, control and 4EGI-1, consisted of 4 female and 4 male UM-HET3 mice aged 6 to 8 months old. Controls received an intraperitoneal injection of 15 μL dimethyl sulfoxide (DMSO) daily for 5 days, and treated mice received DMSO containing 4EGI-1 at 75 mg per kg body weight. After the last injection, the mice were fasted for 18 h prior to euthanasia. Excised livers were washed in PBS and snap-frozen for proteomic analysis. Hereinafter, this experiment is called LC-M004. All the animal protocols in the LC-M001 and LC-M004 experiments were approved by the University of Michigan's Institutional Animal Care and Use Committee (IACUC).

The frozen livers were dissected, processed with lysis and trypsin digestion, and analyzed by mass spectrometry (MS) for quantitative protein abundance. Liver sections were placed in lysis buffer (50 mM tris(hydroxymethyl)aminomethane (Tris)-HCl pH 8.0 and 5% sodium dodecyl sulfate (SDS)) and homogenized using a Precellys® 24 tissue homogenizer (Bertin Technologies SAS, Montigny-le-Bretonneux, France). For each sample, protein concentrations were determined by a bicinchoninic acid (BCA) assay. A total of 300 μg of solubilized protein extract in 5% SDS was purified to remove SDS using Midi-S-Trap™ sample processing technology (ProtiFi, New York, USA), and digested with trypsin at 37 °C for 4 h. The extracted tryptic peptides were subjected to reverse phase liquid chromatography tandem mass spectrometry (LC-MS/MS), using an Easy-nLC 1000 (Thermo Fisher Scientific, Massachusetts, USA) with a 50 cm fused silica capillary (75 μm inner diameter) packed with C18 (ReproSil-Pur 1.9 μm; Dr. Maisch GMBH, Ammerbuch, Germany) heated at 45 °C. The mobile phase gradient consisted of 5–35% acetonitrile and 0.1% formic acid over 3 h for the LC-M001 samples or over 2 h for the LC-M004 samples. The LC-M001 samples were analyzed on a Q Exactive-HF mass spectrometer (Thermo Fisher Scientific) in data-dependent acquisition (DDA) mode with an MS scan mass range of 375–1375 $m/z$ and a resolution of 60,000. MS/MS scans were acquired with TopN = 15 using 15,000 resolution, with an isolation width of 1.8 $m/z$, AGC set to 100,000, and 100 ms injection time. NCE was set to 27, and dynamic exclusion was set to 20 s. The LC-M004 samples were analyzed on an Orbitrap Fusion Lumos (Thermo Fisher Scientific) in DDA mode with an MS scan mass range of 375–1375 $m/z$ and a resolution of 60,000. MS/MS scans were acquired with TopN = 12 using 15,000 resolution with an isolation width of 1.8 $m/z$, AGC set to 40,000, and 30 ms injection time. NCE was set to 30, and a dynamic exclusion was set to 30 s.

MS data analysis was conducted using the Trans-Proteomic Pipeline[70]. Peptide identification was performed by database searching with Comet[71] using the mouse reference proteome UP000000589 (UniProt, downloaded on June 11, 2019) filtered to one protein sequence per gene. Peptide sequences were validated with PeptideProphet[72] and iProphet[73]. Protein inference was performed with ProteinProphet[74]. Protein quantification was performed using the top-3 method[75,76] on quantities obtained from the extracted ion chromatograms of the precursor signals of the identified proteotypic peptides.

**Mouse liver transcriptomic dataset**. The processed dataset of mouse liver transcriptomics was kindly provided by Vadim N. Gladyshev (Harvard Medical School). Complete descriptions are found in the original paper[45]. Briefly, the original experiment (referred as M001-related in the current study) was designed to investigate eight lifespan-extending interventions: two genetically modified models (the growth hormone receptor knockout mouse (GHRKO) and the hypopituitary Snell dwarf mouse (SnellDW)), two nutritional interventions (calorie restriction (CR) and methionine restriction (MR)), and four pharmacological interventions (ACA, 17aE2, Protandim®, and Rapa). Among 78 mice in total, three mice were prepared for each of the sex- and age-distinguished sample groups, but these groups were not necessarily balanced among interventions (i.e., only the male group was prepared for GHRKO, SnellDW, and MR, and only the 6 months-old group was prepared for 17aE2 and Protandim). Based on the sample size for systems analyses, we chose to pool samples per intervention and investigate only the four sample groups (control, ACA, Rapa, and CR) throughout the current study, while acknowledging that this strategy is a study limitation (see Discussion). Each of these four groups consisted of three 6 months-old female, three 12 months-old female, three 6 months-old male, and three 12 months-old male UM-HET3 mice ($n = 12$ mice per group). The interventions were treated via daily feeding of the Purina 5LG6 diet with ACA (1000 mg kg$^{-1}$), Rapa (42 or 14 mg kg$^{-1}$ for 6 or 12 months-old, respectively), or CR (40% less than control) starting at 4 months. The animal protocols were approved by the IACUC at the University of Michigan. The liver samples were processed for paired-end RNA sequencing using NovaSeq 6000 sequencing system (Illumina, California, USA). The

processed reads after the quality filtering and adapter removal were mapped to gene and counted. After filtering out genes with low number of reads, the count data of the filtered genes was passed to the relative log expression (RLE) normalization.

**Data preprocessing**. Considering the small sample size and prioritizing robustness of downstream systems analyses, we chose to pool samples per intervention throughout the current study, while acknowledging the limitation of this approach (see Discussion). Hence, as the first preprocessing step, the potential effects of sex and age were regressed out from each omics dataset. Log-transformed and standardized analyte abundance values were regressed against sex and age with ordinary least squares (OLS) linear regression for each analyte using the Python statsmodels (version 0.13.0) library. Age was included as a categorical variable at the month level, and thus a dummy variable in cases of the LC-M001 and LC-M004 datasets. The residuals were scaled and shifted back to log-scale, and used as the covariate-adjusted analyte abundance values in all further analyses.

**Weighted gene co-expression network analysis**. Weighted Gene Co-expression Network Analysis (WGCNA) was performed for the LC-M001 proteomics, using R WGCNA package (version 1.71) according to the WGCNA methodlogy[47]. Analytes were initially filtered based on missing values with the default threshold setting (50%), and the remaining analytes (3468 proteins) were used to generate the co-expression network. Network generation was performed using Spearman's correlation and the signed-hybrid approach within the WGCNA package. The $\beta$ parameter to approximate a scale-free topology was defined with 8, using the pickSoftThreshold function. Module identification was subsequently performed using the topological overlap matrix and the default hierarchical clustering approach with dynamic tree cut. Consequently, 10 modules were identified from the LC-M001 proteomics (Fig. 3a, Supplementary Data 2). The identified modules were summarized with module eigengene: the $q$-module eigengene $E^{(q)}$ corresponds to the first principal component of the expression matrix of proteins in that module. In addition, intramodular connectivity (i.e., the sum of the adjacency to the other nodes within the module) was calculated for each protein of the modules.

**Differential Rank Conservation analysis — data processing**. To apply Differential Rank Conservation (DIRAC) analysis, missingness in the mouse datasets was conservatively resolved by filtering out the analytes that were not detected in one or more samples; the final number of analytes was 2231 proteins for DIRAC analysis of the LC-M001 proteomics, 2112 proteins for DIRAC analysis of the LC-M001 and LC-M004 proteomics, and 11,326 transcripts for DIRAC analysis of the M001-related transcriptomics. The analyte abundance values were normalized using robust $Z$-score (i.e., $Z$-score using median and median absolute deviation (MAD) instead of mean and s.d., respectively) for each sample, and further normalized using robust $Z$-score for each analyte based on the median and MAD of the control group.

**Differential Rank Conservation analysis — module set preparation**. For each protein in the preprocessed proteomic datasets, the Gene Ontology Biological Process (GOBP) annotations were retrieved using the QuickGO[77] application programming interface (API) with a query of UniProt ID (January 26, 2021). For each gene in the preprocessed transcriptomic dataset, the GOBP annotations were retrieved using R org.Mm.eg.db package (version 3.14.0) with a query of the Ensembl ID. Each GOBP term defines a priori module consisting of all annotated proteins/genes in the corresponding species (i.e., backgrounds). To maintain the biological meaning of annotation, the modules were further selected if at least half of the members in the module, with a minimum of four members, were quantified in the processed datasets; the final a priori module set was 164 modules for DIRAC analysis of the LC-M001 proteomics (Supplementary Data 1), 153 modules for DIRAC analysis of the LC-M001 and LC-M004 proteomics (Supplementary Data 7), and 3912 modules for DIRAC analysis of the M001-related transcriptomics (Supplementary Data 5). Note that, although there are multiple annotation databases, we representatively used GOBP annotations throughout the current study simply based on its largest coverage.

Data-driven modules were prepared by applying WGCNA to the LC-M001 proteomics, as described above. Because missingness was differently handled between DIRAC analysis and WGCNA, each WGCNA-identified module could have the analytes that were not retained in the processed dataset for DIRAC analysis (Fig. 3a). Hence, the WGCNA modules were further selected if at least half of the members in the module, with a minimum of four members, were retained in the preprocessed dataset; the final data-driven module set was six modules for DIRAC analysis of the LC-M001 proteomics (Supplementary Data 4).

**Differential Rank Conservation analysis — DIRAC calculation**. The DIRAC algorithm[44] (Fig. 1b) was reimplemented in Python (version 3.7.6 or 3.9.7). Briefly, pairwise comparisons of analyte values within a module are initially performed for each sample, generating a ranking/ordering dataframe which contains binary values about whether analyte$_i$ value is larger than analyte$_j$ value (i.e., differential ranks). Next, consensus of the binary values is calculated per analyte$_i$–analyte$_j$ pair for each sample group (called phenotype in the original paper) by majority vote, generating a binary ranking/ordering template dataframe which corresponds to the

rank consensus in the DIRAC algorithm. Then, each analyte$_i$–analyte$_j$ pair in the ranking/ordering dataframe is judged whether it matches or mismatches with a consensus in the ranking/ordering template dataframe. Rank matching score (RMS) for each module against each consensus is obtained per sample by calculating a rate of the number of matched pairs. Finally, RMSs for each module against each consensus is summarized with the arithmetic mean per sample group. When this module mean of RMSs in a sample group is based on the consensus of the sample group itself, it corresponds to rank conservation index (RCI); that is, RCI is a special case of the module RMS mean.

**Genome-scale metabolic model reconstruction**. For each of the samples in the M001-related transcriptomics, a context-specific (i.e., sample-specific) metabolic network model was reconstructed from a mouse genome-scale metabolic model (GEM), iMM1865[49], which is a knowledge-based multi-compartment model consisting of 1865 metabolic genes, 10,612 reactions, and 5839 metabolites. According to the gene–protein–reaction (GPR) associations, the analyte abundance values were integrated with the generic iMM1865 for each sample using the integrative metabolic analysis tool (iMAT) algorithm[78]. Subsequently, to predict the flux values of reactions at steady state, flux variability analysis (FVA) was performed for each context-specific GEM using the COBRA toolbox (version 3.0)[79]. FVA evaluates the flux range for each reaction by optimizing all the potential flux distributions to minimize or maximize a pre-defined objective function under the solution space (i.e., under the context-specific constraints), which is known as the LP (Linear Programming) and MILP (Mixed Integer Linear Programming) problems. The biomass reaction (BIOMASS_reaction) defined in the generic iMM1865 was used as the objective function to be maximized, and FVA was performed for 90% of the optimal solution using the fastFVA function. COBRA toolbox was implemented in MATLAB (R2019a), and academic licenses of Gurobi optimizer (version 7.5) and IBM CPLEX (version 12.7.1) were used to solve LP and MILP. As a result, the flux ranges were successfully predicted for 7834 reactions among the 10,612 reactions of the generic GEM. Then, reactions without subsystem annotation (e.g., BIOMASS_reaction) and reactions in the exchange/demand, miscellaneous, or transport subsystems were removed (because they are pseudo-reactions of the GEM system or their functional meaning is difficult to be interpreted as a single subsystem), and only the remaining 3736 functional reactions were assessed (Supplementary Data 8). The average of the minimum and maximum values in FVA was representatively used as the predicted flux value.

**Statistics and reproducibility**. All processing and null hypothesis testing were performed using Python (version 3.9.7) with Python NumPy (version 1.21.3), pandas (version 1.3.4), SciPy (version 1.7.1) and statsmodels (version 0.13.0) libraries, except for overrepresentation analysis using R (version 4.1.1) with R tidyverse (version 1.3.1) and clusterProfiler (version 4.2.2)[80] packages. All statistical tests were performed using a two-sided hypothesis. In all cases of multiple testing, $P$-values were adjusted with the Benjamini–Hochberg method. Statistical significance was based on $P < 0.05$ for single testing and FDR $< 0.05$ for multiple testing. All data were derived from independent mice (i.e., biological replicates). Group statistics (e.g., sample size, mean, s.e.m.) and test summary (e.g., degrees of freedom, test statistic, $P$-value) are found in Supplementary Data 1–9.

Differences in overall RCI distribution between control and each intervention were assessed using Mann–Whitney $U$-tests while adjusting multiple testing across three (Figs. 2a, 3f, 4a) or four (Fig. 5a) comparisons. For identifying the module changed by one or more of the interventions, the intervention effect on RCI (i.e., the mean of RMSs under the rank consensus of own sample group) was assessed using Analysis of Variance (ANOVA; RMS ~ intervention) for each module or for each module and each omics dataset, while adjusting multiple testing across 164 (Fig. 2b, c), six (Fig. 3g), or 3912 (Supplementary Fig. 4a, b) modules or across 312 (156 modules × two omics datasets; Fig. 4b) tests. Note that GOBP modules are partly dependent on each other because the same gene/protein can be shared between GOBP terms; hence, this simple adjustment approach could inflate false negatives, and thus be regarded as a conservative approach. For subsequently clarifying which intervention changed (i.e., tightened or loosened) the module, the post hoc comparisons for RCI between control and each intervention were assessed using Welch's $t$-tests while adjusting multiple testing across three (Figs. 2b–d, 3g, 3h; Supplementary Figs. 2a, 2c, 2e, 2g, 3e, 4a–c) or four (two comparisons × two omics datasets; Figs. 4b–d, 5d, 5g) comparisons. For examining the similarity of module regulation among interventions, differences in the mean of RMSs between control and each intervention were assessed for each rank consensus using Welch's $t$-tests while adjusting multiple testing across six (two comparisons × three rank consensus; Figs. 2c, 2e, 3i; Supplementary Figs. 2b, 2d, 2f, 2h, 3f, 4b, 4d) comparisons. Note that the sample group corresponding to the rank consensus group was excluded in these tests, because its mean of RMSs (i.e., RCI) is expected to follow a different distribution from the other sample groups' one. Also note that, because false negatives were inflated due to double tests (e.g., Control vs. ACA under the Rapa rank consensus was complementary to Control vs. Rapa under the ACA rank consensus), this simple adjustment approach was regarded as a conservative approach. For identifying the module whose regulation similarity to the LC-M004 sample groups (Control-2, 4EGI-1) was different among the LC-

M001 sample groups (Control-1, ACA, 17aE2, Rapa), the intervention effect on the RMS mean was assessed using ANOVA (RMS ~ intervention) for each module and each rank consensus, while adjusting multiple testing across 306 (153 modules × two rank consensus; Fig. 5b, e) tests (i.e., a conservative approach, as described above). For subsequently clarifying which intervention (similarly or dissimilarly) changed the module, the post hoc comparisons for the RMS mean between Control-1 and each intervention (ACA, 17aE2, Rapa) were assessed using Welch's $t$-tests while adjusting multiple testing across six (three comparisons × two rank consensus; Fig. 5b, c, e, f) comparisons.

For identifying the WGCNA module changed by one or more of the interventions, the intervention effect on the module eigengene was assessed using ANOVA ($E^{(q)}$ ~ intervention) for each module, while adjusting multiple testing across 10 modules. For subsequently clarifying which intervention changed the module eigengene, the post hoc comparisons for the $E^{(q)}$ mean between control and each intervention were assessed using Welch's $t$-tests while adjusting multiple testing across three (Fig. 3b, Supplementary Fig. 3a) comparisons. For examining the relationship between the intervention effect on each protein and their respective intramodular connectivity in the Blue (Fig. 3c) or Pink (Supplementary Fig. 3b) module, the main effect of intervention on each protein $k$ was calculated using ANOVA (Protein$_k^{(q)}$ ~ intervention), and then Spearman's correlation between the calculated main effect of intervention and intramodular connectivity was assessed. For investigating characteristics of the Blue or Pink module, enrichment in the top 10% hub proteins (Blue: 79 proteins, Pink: 18 proteins) was assessed using overrepresentation tests for each of the GOBP terms that were mapped to any of the hub proteins, while adjusting multiple testing across 81 (Fig. 3d) or seven (Supplementary Fig. 3c) terms.

For examining the GEM subsystems that were shifted by each intervention, the reactions that were potentially changed by the intervention were first screened for, and then the GEM subsystems that were enriched in them were identified. To screen for the potentially changed reactions, differences in flux value distribution between control and each intervention were assessed using Mann–Whitney $U$-tests for each reaction, and the reactions having nominal $P < 0.05$ were selected. Enrichment in the potentially changed reactions (ACA: 170 reactions, Rapa: 0 reaction, CR: 38 reactions) was assessed using overrepresentation tests for each of the GEM subsystems that were annotated to any of the potentially changed reactions, while adjusting multiple testing across 21 (Fig. 6b) or 11 (Fig. 6c) subsystems. Note that reactions in GEM are partly dependent on each other because the same gene/protein/metabolite can be shared between the reactions; hence, this simple adjustment approach could inflate false negatives, and thus be regarded as a conservative approach.

**Data visualization**. Most results were visualized using Python (version 3.9.7) with Python matplotlib (version 3.4.3), seaborn (version 0.11.2), venn (version 0.1.3) libraries, while the results of enrichment analyses were visualized using R (version 4.1.1) with R ggplot2 (version 3.3.6) and enrichplot (version 1.14.2) packages. The results were summarized as the mean with 95% confidence interval (CI) or the standard boxplot (median: center line; 95% CI around median: notch; [$Q_1$, $Q_3$]: box limits; [$x_{min}$, $x_{max}$]: whiskers, where $Q_1$ and $Q_3$ are the 1st and 3rd quartile values, and $x_{min}$ and $x_{max}$ are the minimum and maximum values in [$Q_1 - 1.5 \times$ IQR, $Q_3 + 1.5 \times$ IQR] (IQR, interquartile range, $Q_3 - Q_1$), respectively), as indicated in each figure legend. Note that this 95% CI of mean or median was simultaneously calculated during visualization using the seaborn barplot or boxplot API, respectively; hence, this CI is not exactly the same as that used in statistical analysis but for presentation purposes only. Hierarchical clustering was simultaneously performed during visualization using seaborn clustermap API with the Ward's linkage method for Euclidean distance. For the values used in Fig. 6a, the group mean of flux values for each reaction was centered by subtracting the group mean of the corresponding control, and then scaled by the maximum absolute value across intervention groups using MaxAbsScaler of Python scikit-learn library (version 1.0.1).

**Reporting summary**. Further information on research design is available in the Nature Portfolio Reporting Summary linked to this article.

## Data availability

The MS data of the LC-M001 and LC-M004 proteomics have been deposited to the ProteomeXchange Consortium via the PRIDE partner repository (PXD035255)[81,82]. The processed data of the M001-related transcriptomics was kindly provided by Vadim N. Gladyshev (Harvard Medical School), and raw data is available on the NCBI's Gene Expression Omnibus (GEO) repository (GSE131901)[83]. Source Data are provided with this paper (Supplementary Data 9).

## Code availability

Code used in this study is freely available on GitHub (https://github.com/longevity-consortium/SysBioM001Paper)[84].

# ARTICLE

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

## Acknowledgements

We thank Vadim N. Gladyshev (Harvard Medical School) for kindly providing the processed dataset of mouse liver transcriptomics; Eric S. Orwoll (Oregon Health and Science University) and Gilbert S. Omenn (University of Michigan) for providing comments to the manuscript; Jennifer Dougherty and Mary Brunkow (Institute for Systems Biology) for their coordination efforts in the National Institute on Aging (NIA) Longevity Consortium. This work was funded by the National Institutes of Health (NIH) grant U19AG023122 awarded by NIA (to J.L., S.R.C., N.J.S., R.A.M., R.L.M., and N.R.) and a generous gift from K. Carole Ellison (to K.W. and T.W.). K.W. was supported by The Uehara Memorial Foundation (Overseas Postdoctoral Fellowships).

## Author contributions

K.W., T.W., P.B., M.R., R.A.M., R.L.M., and N.R. conceptualized the study. K.W., T.W., P.B., M.R., R.A.M., and N.R. participated in the study design. G.G.G. and R.A.M. performed mouse experiments. M.R.H., M.K.M., D.H.B., M.M., S.R.M., K.M.C., C.K., U.K., and R.L.M. contributed to the generation of the mouse proteomics. K.W., T.W., and P.B. performed data analysis and figure generation. J.W., J.L., L.P., C.L., J.C.R., G.G., S.R.C., N.J.S., N.D.P., and L.H. assisted in result interpretation. K.W., T.W., P.B., M.R., and N.R. were the primary authors of the paper, with contributions from all other authors. All authors read and approved the final manuscript.

## Competing interests

The authors declare no competing interests.
