## [Peer Review File · Communications Biology]

Reviewers' comments:

Reviewer #1 (Remarks to the Author):

The paper used differential rank conversation (DIRAC) analysis to investigate the regulation tightening of biological modules under lifespan-extending interventions. The statistical analysis is satisfactory overall. Biological discoveries, such as the tightened regulation of fatty acid metabolism and inflammation processes, are intriguing. My specific comments are as below.

Major:

- Line 583: would it be better to regress out sex, age, and other cofounders when computing the z-scores?

- Figure 2: what's the correlation of RCI across modules between the 4 conditions?

- Fig. 3a: maybe assign a name to each data-driven module (e.g., lead gene) instead of using color names.

Reviewer #2 (Remarks to the Author):

On a high level, I feel that this is a well-designed, sophisticated, and thoughtful manuscript. The approach is innovative, the technological approach is sound, and the findings are likely to be of interest to the biogerontological community. I do, however, have some minor recommendations for improvement. My specific comments are as follows:

- In the first paragraph of the Introduction, classical pathways pertinent to aging (i.e., mTOR, AMPK, sirtuins, and insulin/IGF-1) are mentioned. Given that the effects of these pathways can be tissue-specific, can the tissue-specificity of pro-longevity interventions be conveyed to better inform the non-expert reader? Using mice as an example, impairing insulin signaling in extends lifespan in adipose tissue (PMID: 12543978) but does not influence longevity in peripheral, non-neuronal tissues (PMID: 28544360).

- In the second paragraph of the Introduction, the authors introduce the term "healthspan" before defining it. Since *Communications Biology* is a broad journal, can the authors explicitly define healthspan? In my opinion, Dr. Matt Kaeberlein has done an especially good job defining this concept (PMID: 30084059). It would also be reasonable to explicitly define "geroprotector".

- In paragraph #3 of the Introduction, the authors state the following: "Rapa is the only drug found to prolong lifespan in every organism studied, including yeast, worms, flies, and mammals." My recommendation is to change this to "Rapa is one of the only drugs..." or convey that you're only referring to molecules tested by the ITP. Molecules like alpha-ketoglutarate and spermidine, for example, have been reported to extend lifespan in multiple model organisms.

- Is there a reason that the authors only looked at GO BP and did not look at other databases such as GO Molecular Function, GO Cellular Component, Reactome, KEGG, Panther, or WikiPathways? As has been demonstrated by others (e.g., PMID: 32311500), enrichment results may differ across distinct databases. While I don't think this is necessary for the current manuscript, I would encourage the authors to provide an explanation for why they honed in on GO BP.

- The authors have performed multiple analyses and generated quite a bit of data. If feasible, a table summarizing the primary findings and key pathways implicated would be helpful.

- Instead of using the term "anti-aging" twice in the Discussion, my recommendation would be to instead use "pro-longevity".

- Typically, a Discussion section ends with high-level recap and/or forward-looking statements. Instead of ending on study limitations, please provide a more typical ending to this section.

- Can the authors provide more insight into what their findings reveal for the biology of aging and the relative importance of established aging hallmarks (PMID: 23746838)?

- In the Code Availability section, the hyperlink does not match the text (which is correct) and is not a page on Github. Please update with the correct link.

Reviewer #3 (Remarks to the Author):

In this study, the authors generate mouse liver proteomic datasets from life-extending interventions which they use, together with a published transcriptomic dataset, to perform various systems-level analyses. They observe consistent changes between different life extending interventions, which I found interesting, and a tightening regulation of processes such as fatty acid metabolism and inflammation. This is a timely and interesting work, though some aspects were unclear to me, as detailed below.

1) One key approach of the study is based on identifying tightened modules due to life extending interventions. However, I would expect that the majority of interventions and treatments (whether they extend lifespan or not) will induce changes and therefore result in a tightening of expression profiles. Intuitively that seems to me to be an expected response to any type of intervention that elicits biological changes, but perhaps I'm missing something in which case I would encourage the authors to clarify this.

2) The authors pooled males and females, yet sex specific effects have been common in the ITP, including from recent genetic analysis (eg, <https://pubmed.ncbi.nlm.nih.gov/36173858/> and <https://pubmed.ncbi.nlm.nih.gov/36173862/>). So I am a bit puzzled why the authors did not at least explore sex specific effects in their data as it could be an opportunity to gain insights on the sex specific effects of the interventions. Again, am I missing something?

3) In the abstract and elsewhere the authors mention "aging-related modules", but as far as I can tell no analysis of aged animals was performed, so I don't understand what the authors mean by "aging-related" in this context.

4) The results are interesting, but at times a little unclear what it all means. For example, in line 414 the authors write that: fatty acid metabolism "emerged as a common process shifted" - what do you mean by "shifted"? Is there an activation of fatty acid metabolism? I think I understand at the computational level how the analysis work, but is not entirely clear to me what it means at the biological level.

5) Line 423: although I think there are many commonalities between cancer and anti-aging (eg see <https://pubmed.ncbi.nlm.nih.gov/23612461/>), I am not fully persuaded by the arguments of the authors because cancer involves so many expression changes, driven by genomic instability, that it is difficult to infer transcriptional regulation in cancer. During aging there are relatively modest gene expression changes (eg see <https://pubmed.ncbi.nlm.nih.gov/19189975/> and <https://pubmed.ncbi.nlm.nih.gov/33611312/>), so at the transcriptional level I don't really find cancer and aging to be equivalent.

6) Although the results are interesting and I applaud the authors for using proteomics, most of the processes identified have been previously found in other studies of longevity and aging. For instance, common signatures of caloric restriction show changes in fatty acid metabolism, cholesterol, peroxisomes and immune responses: <https://pubmed.ncbi.nlm.nih.gov/22327899/>
As such, I think the perhaps the authors wish to emphasize the novel aspects of their work.

Overall, I think this is an interesting work, but some aspects I think could be improved. It is possible that some of my comments reflect misunderstandings of mine. If so then I would suggest that the authors use my misunderstandings as an indication that such points might be made clearer in the manuscript.

It is my usual policy to reveal my identity to the authors: Joao Pedro de Magalhaes.

Point-By-Point Response

We would like to thank the reviewers for their constructive comments, which helped us to drastically improve the clarity and quality of our manuscript. We have submitted a revised version of the manuscript, in which all changes from the initial submission are highlighted with red-colored text and the points that are directly related to the reviewers' comments are highlighted with yellow marker.

To address the excellent suggestion put forward by Reviewer #1, we have regressed out potential confounders from all used datasets prior to performing our systems analyses. This change in preprocessing required for re-performing all analyses and thus re-generating all figures and tables. The re-analyses of Differential Rank Conservation (DIRAC) and Weighted Gene Co-expression Network Analysis (WGCNA) yielded additional findings compared to the previous results. This is likely due to the improved statistical power achieved by eliminating covariate-related variations within each sample group. In contrast, the re-analysis of genome-scale metabolic model (GEM) modified the findings, specifically for methionine restriction (MR), growth hormone receptor knockout (GHRKO), and Snell dwarf (SnellDW) groups. Despite our initial concern about the small sample size of MR, GHRKO, and SnellDW groups ($n = 3$ mice per group), we included all intervention groups of the original study (Tyshkovskiy, A. et al. *Cell Metab.* 2019) in our previous analysis (because we did see the statistical significance even under a conservative approach). Given the newer results, however, we cannot deny the possibility that the previous GEM findings about MR, GHRKO, and SnellDW were artifacts derived from covariate-related variations. Note that, even for the “negative” results (e.g., reaction that was not changed with statistical significance) in the newer results, one cannot conclude them as the “negative” findings (e.g., the reaction is not affected by the intervention). Hence, during the current revision, we decided to change our analysis strategy to focus only on the three interventions (acarbose (ACA), rapamycin (Rapa), and calorie restriction (CR)), which had sufficient sample size ($n = 12$ mice per group). Note that this strategy is the same with the DIRAC analysis of this dataset. Therefore, the revised version of the manuscript contains major changes compared to the previous version. Nevertheless, there is no material change to the main findings or conclusions of the paper.

Further edits were made to enhance the clarity of our manuscript for both expert and non-expert audiences. We are confident that the revised version of the manuscript will appeal to the broad readership of *Communications Biology*.

In the subsequent responses, we have considered and addressed all comments provided by each reviewer. Text colored with light blue represents the original reviewers' comments.

Reviewer #1:

Remarks to the Author:

The paper used differential rank conversation (DIRAC) analysis to investigate the regulation tightening of biological modules under lifespan-extending interventions. The statistical analysis is satisfactory overall. Biological discoveries, such as the tightened regulation of fatty acid metabolism and inflammation processes, are intriguing. My specific comments are as below.

We are grateful for the positive assessment of the reviewer.

Major:

- Line 583: would it be better to regress out sex, age, and other cofounders when computing the z-scores?

We appreciate the reviewer's excellent suggestion. All the datasets used in this study were obtained from mouse experiments, where potential confounders were controlled as much as possible during the experimental design and data generation. For instance, mice were randomly assigned into control and intervention groups from the same mouse colony, and each sample size was balanced between sexes. Moreover, liver samples were randomly assigned into plate wells during mass spectrometry (MS) measurement, and technical biases were adjusted during MS data analysis. Therefore, in contrast to general observational studies, the effects of potential confounders on results were expected to be minor and thus able to be ignored, which is generally assumed in experimental studies (e.g., mouse phenotype measurements, cellular phenotype measurements, molecular function measurements).

Nevertheless, we agree that it is better to adjust potential confounders as much as possible, specifically in our case where the effects of the lifespan-extending interventions may be confounded by sex. During the current revision, we regressed out the available potential covariates (i.e., sex and age) from all the datasets as the first step of preprocessing, and re-performed all the analyses using these adjusted values.

The DIRAC and WGCNA re-analyses successfully reproduced the previous findings (e.g., fatty acid oxidation was significantly tightened by the lifespan-extending interventions; a WGCNA-identified module having mitochondrial proteins as its hubs was significantly changed by 17aE2 and Rapa but not by ACA). Moreover, these newer analyses yielded additional findings (e.g., stress granule assembly was significantly tightened by the lifespan-extending interventions; a WGCNA-identified module having ribosomal proteins as its hubs was significantly changed by 17aE2), probably because the statistical power was improved by the removal of sex and age-related variations within each sample group.

In contrast, the GEM re-analysis modified the findings, specifically for MR, GHRKO, and SnellDW groups. Despite our initial concern about the small sample size of MR, GHRKO, and SnellDW groups ($n = 3$ mice per group), we included all intervention groups of the original study (Tyshkovskiy, A. et al. *Cell Metab.* 2019) in our previous analysis (because we did see the statistical significance even under a conservative approach). Given the contradiction between the previous and revised results, one may optimistically consider that the adjusted covariates (i.e., sex and age) were associated with the true effects of lifespan-extending interventions and thus regressing them out was too conservative. However, we cannot deny the possibility that the previous GEM findings about MR, GHRKO, and SnellDW were artifacts derived from sex or age-related variations. Note that, even for the "negative" results (e.g., reaction that was not changed with statistical significance) in the revised results, one cannot conclude them as the "negative" findings (e.g., the reaction is not affected by the intervention). Hence, during the current revision, we decided to change our analysis strategy to focus only on the three interventions (ACA, Rapa, and CR), which had sufficient sample size ($n = 12$ mice per group). Note that this strategy is the same with the DIRAC analysis of this dataset. Although it is unknown whether fatty acid oxidation was shifted by MR, GHRKO, and SnellDW, the revised analysis revealed that it was significantly shifted by ACA and CR at least.

Therefore, the newer results contain major changes compared to the previous results in result details, but there is no material change to the main findings or conclusions: (1) mechanistically distinct lifespan-extending interventions generally tightened the systems-level profiles of biological processes; (2) fatty acid metabolism emerged as a common process shifted by multiple lifespan-extending interventions; (3) lifespan-extending interventions achieved the tight proteomic profiles of biological processes through transcriptional or post-transcriptional regulation, potentially including CIT. In the current revision, we replaced all the

previous results with the revised ones, and revised texts while clearly mentioning the adjustments of sex and age within the main body (lines 132–134 of page 3, lines 262–266 of page 6, lines 368–370 of page 8) in addition to the Method section. We thank the reviewer for providing the opportunity for us to report more robust results.

- Figure 2: what's the correlation of RCI across modules between the 4 conditions?

We appreciate the reviewer's thoughtful comment. We agree that Fig. 2a lacks the information of module difference (i.e., Fig. 2a could be obtained even if the set of modules showing higher rank conservation index (RCI) was completely different between the intervention groups), and that presenting the pairwise correlations of RCIs among groups would be an alternative way. That is, if one observes high RCI correlations between the intervention groups but low correlations with control group, it would become a better presentation for highlighting the commonality among the interventions. However, as shown in Fig. 2b, some modules still exhibited higher RCIs specifically for an intervention, and thus we concerned that the interpretation of overall correlations might not be simple. In fact, that was the case; i.e., a significant RCI correlation was observed in 17aE2 vs. Rapa and ACA vs. 17aE2 but also in Control vs. Rapa despite not in Aca vs. Rapa (Point-By-Point Response Fig. 1). This would be mainly because the modules having higher RCIs specifically for an intervention hinder the overall correlation. Therefore, we think a correlation plot is less useful to show the overall RCI distribution in this case, and we chose to keep the initial presentations (Fig 2a and Fig. 2b).

Point-By-Point Response Fig. 1. Pairwise correlations of RCI among Control, ACA, 17aE2, and Rapa. RCI values were derived from DIRAC results of the LC-M001 proteomics data using Gene Ontology Biological Process (GOBP)-defined modules, corresponding to Fig. 2. P_{adj} : P -value in Spearman's correlation test after the Benjamini–Hochberg adjustment across six pairs. $n = 164$ GOBP modules.

- Fig. 3a: maybe assign a name to each data-driven module (e.g., lead gene) instead of using color names.

We appreciate the reviewer's constructive suggestion. The reason of using color code for WGCNA module was based on the default setting of WGCNA package, and thus the color code name may be rather comfortable for readers who are familiar with WGCNA. Using the top hub protein for the name of WGCNA module may be useful for some readers but also raise a risk of

confusion to other readers as if we compared concentrations of the protein. Hence, we chose to retain the color code for the name of WGCNA module.

To still address the reviewer's comment, we added an enrichment analysis on the top hub proteins (Fig. 3d, Supplementary Fig. 2c), and integrated them into the main body for readers to capture the characterization of WGCNA modules (lines 211–214 and 219–221 of page 5). Moreover, to support readers' understanding, the characterized feature was added as an adjective: e.g., the mitochondrial Blue module (line 233 of page 5) and the ribosomal Pink module (line 243 of page 5).

Reviewer #2:

Remarks to the Author:

On a high level, I feel that this is a well-designed, sophisticated, and thoughtful manuscript. The approach is innovative, the technological approach is sound, and the findings are likely to be of interest to the biogerontological community. I do, however, have some minor recommendations for improvement. My specific comments are as follows:

We are grateful for the positive assessment of the reviewer.

- In the first paragraph of the Introduction, classical pathways pertinent to aging (i.e., mTOR, AMPK, sirtuins, and insulin/IGF-1) are mentioned. Given that the effects of these pathways can be tissue-specific, can the tissue-specificity of pro-longevity interventions be conveyed to better inform the non-expert reader? Using mice as an example, impairing insulin signaling in extends lifespan in adipose tissue (PMID: 12543978) but does not influence longevity in peripheral, non-neuronal tissues (PMID: 28544360).

We appreciate the reviewer's constructive suggestion. We agree that the tissue specificity is to be mentioned here, because it supports the necessity of systems-level approach. In the revised manuscript, we added this point into the background of Introduction section (lines 60–61 of page 2).

- In the second paragraph of the Introduction, the authors introduce the term “healthspan” before defining it. Since *Communications Biology* is a broad journal, can the authors explicitly define healthspan? In my opinion, Dr. Matt Kaeberlein has done an especially good job defining this concept (PMID: 30084059). It would also be reasonable to explicitly define “geroprotector”.

We appreciate the reviewer's suggestion. We agree that these words could be jargons for non-expert readers. In the revised version, we defined these terms and cited the suitable papers (lines 66, 68–69 of page 2).

- In paragraph #3 of the Introduction, the authors state the following: “Rapa is the only drug found to prolong lifespan in every organism studied, including yeast, worms, flies, and mammals.” My recommendation is to change this to “Rapa is one of the only drugs...” or convey that you're only referring to molecules tested by the ITP. Molecules like alpha-ketoglutarate and spermidine, for example, have been reported to extend lifespan in multiple model organisms.

We appreciate the reviewer's observation regarding the problematic usage of the phrase “the

only drug". We have made the necessary correction in response to this feedback (line 73 of page 2).

- Is there a reason that the authors only looked at GO BP and did not look at other databases such as GO Molecular Function, GO Cellular Component, Reactome, KEGG, Panther, or WikiPathways? As has been demonstrated by others (e.g., PMID: 32311500), enrichment results may differ across distinct databases. While I don't think this is necessary for the current manuscript, I would encourage the authors to provide an explanation for why they honed in on GO BP.

We appreciate the reviewer's reasonable suggestion. The reason why we used GOBP throughout this study was simply based on its largest coverage. In the revised manuscript, we mentioned this point in the Method section (lines 598–600 of page 13).

- The authors have performed multiple analyses and generated quite a bit of data. If feasible, a table summarizing the primary findings and key pathways implicated would be helpful.

We appreciate the reviewer's constructive suggestion. We originally intended to present primary findings and key pathways as Figures and to provide comprehensive pathway lists as Supplementary Data tables. While considering the list length, we added a key list of the LC-M001 proteomics analysis as the main table (Table 1).

- Instead of using the term "anti-aging" twice in the Discussion, my recommendation would be to instead use "pro-longevity".

We appreciate the reviewer's comment. Actually, the primary authors used the term pro-longevity in the earlier drafting stage. However, there was a discussion about this terminology even among the co-authors. The term pro-longevity is a jargon, which was originally introduced in the 1960's by fringe characters in the immortality movement and designed to be "cute" in that the "long" syllable is ambiguously part of both "prolong" and "longevity." Therefore, we decided to avoid the use of this term throughout the manuscript.

- Typically, a Discussion section ends with high-level recap and/or forward-looking statements. Instead of ending on study limitations, please provide a more typical ending to this section.

We appreciate the reviewer's kind suggestion. We added a high-level summary in the last paragraph of Discussion (lines 475–478 of page 10).

- Can the authors provide more insight into what their findings reveal for the biology of aging and the relative importance of established aging hallmarks (PMID: 23746838)?

We appreciate the reviewer's comment. In the previous version, we intended to mention our findings in the context of the known concepts such as "aging hallmarks" and "pillars of aging", by utilizing their citations. However, we agree with the reviewer that this was not sufficient. In the revised version, we improved the Discussion section to clearly describe why our findings are important in the biology of aging compared to the previous knowledge (lines 411–416, 421–425 of page 9).

- In the Code Availability section, the hyperlink does not match the text (which is correct) and is not a page on Github. Please update with the correct link.

We thank the reviewer for bringing this to our attention. This seemed to have happened due to the conversion from word file to PDF file. We will make sure that this does not happen during proofing.

Reviewer #3:

Remarks to the Author:

In this study, the authors generate mouse liver proteomic datasets from life-extending interventions which they use, together with a published transcriptomic dataset, to perform various systems-level analyses. They observe consistent changes between different life extending interventions, which I found interesting, and a tightening regulation of processes such as fatty acid metabolism and inflammation. This is a timely and interesting work, though some aspects were unclear to me, as detailed below.

We are grateful for the positive assessment of the reviewer.

1) One key approach of the study is based on identifying tightened modules due to life extending interventions. However, I would expect that the majority of interventions and treatments (whether they extend lifespan or not) will induce changes and therefore result in a tightening of expression profiles. Intuitively that seems to me to be an expected response to any type of intervention that elicits biological changes, but perhaps I'm missing something in which case I would encourage the authors to clarify this.

We appreciate the reviewer's thoughtful comment. Indeed, one can intuitively hypothesize that treatment by any drugs would change expression profiles toward tightening in general. Indeed, it is our findings that confirm this hypothesis in a rigorous and quantifiable way. Moreover, the tightening response does not necessarily have to be universal for all modules/systems. In fact, we observed some modules that did not change by the three lifespan-extending drugs (Fig. 2b). In addition, although module tightening was observed in the current study, the original DIRAC paper (Eddy, J. A. et al. *PLoS Comput. Biol.* 2010) revealed module loosening in the context of cancer stage. This inverse pattern between longevity and cancer severity leads to an interesting insight, as described in Discussion (lines 411–416 of page 9). Moreover, the primary point we make in the paper is that multiple lifespan-extending drugs, whose primary modes of action are different, exhibited similar tightening patterns for some modules (e.g., Fig. 2e). This ability to compare the tightening patterns between interventions is one of the advantages for using DIRAC analysis. Therefore, we believe that our findings represent a significant scientific advancement.

2) The authors pooled males and females, yet sex specific effects have been common in the ITP, including from recent genetic analysis (eg, <https://pubmed.ncbi.nlm.nih.gov/36173858/> and <https://pubmed.ncbi.nlm.nih.gov/36173862/>). So I am a bit puzzled why the authors did not at least explore sex specific effects in their data as it could be an opportunity to gain insights on the sex specific effects of the interventions. Again, am I missing something?

We appreciate the reviewer's thoughtful comment. We completely agree that sex-specific effects

need to be considered for lifespan-extending interventions, as highlighted in our Introduction section (lines 79–86 of page 2). When we stratified samples with intervention–sex and performed DIRAC analysis, we observed the general tightening by the lifespan-extending drugs in both female and male mice (Point-By-Point Response Fig. 2), consistent with the pooled analysis (Fig. 2a). However, while considering the limited power of our datasets to detect the potential sex-specific effects, we took the strategy of pooling data from both sexes throughout the current study to increase the statistical power and precision of the downstream analyses, leading to more robust findings and more reliable conclusions. We do acknowledge that this is a major limitation of our datasets, and plan to conduct stratified studies when larger datasets become available in the NIA Longevity Consortium. This limitation is clearly described in the Result, Discussion, and Method sections (e.g., lines 132–134 of page 3, lines 262–266 of page 6, lines 467–469 and lines 471–474 of page 10, lines 552–554 of page 12). Moreover, the current revision improved our strategy by regressing out the potential effects of sex as an additional preprocessing step prior to all analyses. Hence, we consider that the current study is worth publishing as a proceeding report about robust effects of lifespan-extending interventions.

Point-By-Point Response Fig. 2. Overall distribution of module RCI. Rank conservation index (RCI) values were derived from DIRAC analysis of the LC-M001 proteomics data using Gene Ontology Biological Process (GOBP)-defined modules, corresponding to Fig. 2. Cont: control, Acar: acarbose, Estr: 17 α -estradiol, Rapa: rapamycin, F: female, M: male. $n = 164$ GOBP modules.

3) In the abstract and elsewhere the authors mention "aging-related modules", but as far as I can tell no analysis of aged animals was performed, so I don't understand what the authors mean by "aging-related" in this context.

We appreciate the reviewer's comment. Our original intention was referring to the modules that were reported in the aging context by previous studies. However, based on the reviewer's comment, we eliminated this unnecessary word (e.g., line 41 of page 1, line 115 of page 2), and improved the discussion about our findings compared to previous studies (lines 421–425 of page 9)

4) The results are interesting, but at times a little unclear what it all means. For example, in line 414 the authors write that: fatty acid metabolism "emerged as a common process shifted" - what do you mean by "shifted"? Is there an activation of fatty acid metabolism? I think I understand at the computational level how the analysis work, but is not entirely clear to me what it means at the biological level.

We appreciate the reviewer's thoughtful comment. We agree that "shifted" is somewhat ambiguous. However, we intentionally used it, because our systems-level results do not indicate the functional direction for cells (e.g., tight regulation can be either augmentation or attenuation of the pathway) (lines 447–449 of page 9). Hence, while citing other studies, we discussed the

biological meaning of this shift in fatty acid oxidation (lines 456–457 of pages 9–10).

5) Line 423: although I think there are many commonalities between cancer and anti-aging (eg see <https://pubmed.ncbi.nlm.nih.gov/23612461/>), I am not fully persuaded by the arguments of the authors because cancer involves so many expression changes, driven by genomic instability, that it is difficult to infer transcriptional regulation in cancer. During aging there are relatively modest gene expression changes (eg see <https://pubmed.ncbi.nlm.nih.gov/19189975/> and <https://pubmed.ncbi.nlm.nih.gov/33611312/>), so at the transcriptional level I don't really find cancer and aging to be equivalent.

We appreciate the reviewer's perspective. We do not claim that cancer and aging processes are equivalent. In addition, our findings were derived from not only transcriptomics (Fig. 5a) but also proteomics (Fig. 2a, 3f), which extends beyond transcriptional changes. In lines 411–416 of page 9, we simply discuss the interesting similarities between cancer resistance and anti-aging from the perspectives of overall systematic patterns (i.e., DIRAC metrics). To avoid misunderstanding, we replaced “commonalities” with “some similarities” in the revised version (line 413 of page 9).

6) Although the results are interesting and I applaud the authors for using proteomics, most of the processes identified have been previously found in other studies of longevity and aging. For instance, common signatures of caloric restriction show changes in fatty acid metabolism, cholesterol, peroxisomes and immune responses:

<https://pubmed.ncbi.nlm.nih.gov/22327899/>

As such, I think the perhaps the authors wish to emphasize the novel aspects of their work.

We appreciate the reviewer's suggestion. We agree that the identified processes themselves (e.g., fatty acid oxidation) confirm results from previous aging and longevity studies, and have been investigated by single molecule-level analyses. However, our study unifies the previously obscure interrelationships between different lifespan-extending interventions into a more coherent framework. Moreover, our study highlights the power of systems-level approaches for identifying and characterizing the biological processes involved in aging and longevity, which can be regarded as a proof-of-concept study expanding on future studies (related to the above point #2). In the revised manuscript, we added the scientific advancement along with a high-level summary of the current study (lines 421–425 of page 9, lines 475–478 of page 10).

Overall, I think this is an interesting work, but some aspects I think could be improved. It is possible that some of my comments reflect misunderstandings of mine. If so then I would suggest that the authors use my misunderstandings as an indication that such points might be made clearer in the manuscript.

It is my usual policy to reveal my identity to the authors: Joao Pedro de Magalhaes.

Thank you Dr. de Magalhães for intently reviewing our manuscript. All the comments were helpful for significantly improving our manuscript.

REVIEWERS' COMMENTS:

Reviewer #1 (Remarks to the Author):

I thank the authors for the extensive follow-up analyses and careful response. Most of my concerns have been adequately addressed. I only have one minor comment, detailed below.

- Figure 2: what's the correlation of RCI across modules between the 4 conditions?

RV1: thank you for the additional analysis. I agree that the initial presentations look better and should be kept. But the correlation scatter plots are also interesting and tell us which interventions induce more similar consequences. I suggest adding them to the supplementary materials.

Reviewer #2 (Remarks to the Author):

The authors have satisfactorily addressed my comments.

Reviewer #3 (Remarks to the Author):

The authors have successfully addressed my concerns and I have no further comments.

Point-By-Point Response

In the following responses, text colored with light blue represents the original reviewers' comments.

Reviewer #1:

Remarks to the Author:

I thank the authors for the extensive follow-up analyses and careful response. Most of my concerns have been adequately addressed. I only have one minor comment, detailed below.

We thank the reviewer for reviewing our revised manuscript and responses.

- Figure 2: what's the correlation of RCI across modules between the 4 conditions?

RV1: thank you for the additional analysis. I agree that the initial presentations look better and should be kept. But the correlation scatter plots are also interesting and tell us which interventions induce more similar consequences. I suggest adding them to the supplementary materials.

Following the suggestion, we provided the correlation plot as Supplementary Figure 1b, and cited in the main text.

Reviewer #2:

Remarks to the Author:

The authors have satisfactorily addressed my comments.

We thank the reviewer for reviewing our revised manuscript and responses.

Reviewer #3:

Remarks to the Author:

The authors have successfully addressed my concerns and I have no further comments.

We thank the reviewer for reviewing our revised manuscript and responses.